# Nanobioconjugates for Signal Amplification in Electrochemical Biosensing

**DOI:** 10.3390/molecules25153542

**Published:** 2020-08-03

**Authors:** Sebastian Cajigas, Jahir Orozco

**Affiliations:** Max Planck Tandem Group in Nanobioengineering, University of Antioquia, Complejo Ruta N, Calle 67 N° 52–20, Medellín 050010, Colombia; sebascajigas06@gmail.com

**Keywords:** nanobioconjugates, signal amplification, electrochemical biosensing, nanomaterials, biomolecules

## Abstract

Nanobioconjugates are hybrid materials that result from the coalescence of biomolecules and nanomaterials. They have emerged as a strategy to amplify the signal response in the biosensor field with the potential to enhance the sensitivity and detection limits of analytical assays. This critical review collects a myriad of strategies for the development of nanobioconjugates based on the conjugation of proteins, antibodies, carbohydrates, and DNA/RNA with noble metals, quantum dots, carbon- and magnetic-based nanomaterials, polymers, and complexes. It first discusses nanobioconjugates assembly and characterization to focus on the strategies to amplify a biorecognition event in biosensing, including molecular-, enzymatic-, and electroactive complex-based approaches. It provides some examples, current challenges, and future perspectives of nanobioconjugates for the amplification of signals in electrochemical biosensing.

## 1. Introduction

Nanochemistry is an emerging research field in the border of chemistry and nanoscience that involves a cross-disciplinary convergence of physical, biological, material sciences, and engineering for multiple purposes [1]. For example, nanochemistry-based nano-bioplatforms exploit natural biomimetic systems that connect chemical and biological systems with nanotechnology and study how such converging technologies help to achieve a better understanding of phenomena at the nanoscale [2,3,4]. With the advent of these emerging research fields, it has been possible to rapidly advance nanobioconjugate architectures with potential in a myriad of cross-disciplinary practical applications [5,6,7,8]. The development of nano-bio-constructs involves several chemical modifications on nanometer scaled structures, of which the properties are size, shape, self-assembly and defects dependent, while features are explained and well-studied in the fields of nanobiotechnology and nanochemistry.

Over the past years, nanobioconjugates have been probed as drug delivery systems, imaging and contrast agents, theranostic platforms, and for purification and concentration of biomolecules. They have received considerable attention in biosensing systems as amplifiers of the resultant signal. Nanobioconjugates are hybrid nano(bio)materials that result from the integration among nanomaterials (NMs) and biomolecules [9]. The development of these hybrid systems aims to get new materials with improved properties concerning each of their components acting alone [1,10]. Each component contributes to the hybrid with a unique property or function that is missing in the other one. Nanobioconjugates are having a significant impact on the development of theranostic (therapeutic and diagnostic) tools in the biomedicine field [1,11,12,13,14,15,16] and have particular potential to revolutionize diagnostic approaches. Limitations of the nanobioconjugates are related to the uncontrolled number of affinity biomolecules that can be linked to the nanostructure, which leads to a variety of unwished phenomena [17]. For instance, a large number of biomolecules linked to nanostructures leads to hindered biochemical activity, alteration of the targeting and biomolecule properties and receptor cross-link [17,18]. Conversely, undesirable multivalent interactions and associated cooperative binding become insignificant if nanostructures have a proper number of linked biomolecules [17].

Within the nanobioconjugates development process, there are some features of the nanomaterials synthesis and the biomolecules coupling that must be controlled to reach stable hybrid materials. The nature of the precursors, their interactions and reagents ratio, the use of surfactants, temperature, time and stirring conditions, among other parameters of the synthesis process, impact on nanomaterial characteristics and their preferential crystal growth [19]. Nanomaterials may present defects depending on the synthesis method, including non-uniform surfaces, edges, lattices, and vertices that serve as points for the enhanced catalysis and anchoring of biomolecules. Bioconjugation employs many nanochemistry-based approaches. For example, carboxylic groups from graphene oxide (GO) can bind covalently aminated biomolecules [1,20,21]. Different crystalline orientations of gold nanosurfaces chemisorb thiol groups naturally from proteins, peptides, DNA strands, and alkyl organic compounds to form self-assembled monolayers (SAMs), among many other examples of bioconjugation [22]. Figure 1A highlights eight different nanobioconjugate configurations [1,10,21]. In the simplest format, the biomolecules might interact directly with the nanostructures (NSs). In more complex formats, the NSs surround or encapsulate the biological components of the nanobioconjugates. Target refers to the (bio)molecule of interest. Bioreceptors, also named ligands, are any kind of molecule that binds to a specific target commonly linked to the transducer, NS, or NM surfaces. Core and shell layers refer to the inner and outermost parts of NSs or NMs. The latter one is commonly bioconjugated with ligands when developing nanobioconjugates and assembling biosensors [1,10,21]. Figure 1B represents a NP decorated with diverse functional biomolecules such as nucleic acids, proteins, drugs, peptides, antibodies, enzymes and others [1,21]. Choosing the bioconjugation strategy is important and depends not only on nanomaterial composition, structure, and available functional groups but on the type of biological molecule, its size, chemical composition, and the requirements of the final application. Figure 1C shows four general conjugation strategies to link biomolecules to NM surfaces [21,23]. Pre- and post-bioconjugation physicochemical metrics include NM size, morphology and aspect ratio, aggregation/agglomeration state, purity, chemical composition, surface characteristics, ζ potential, surface area, and stability, as well as solubility, structure, orientation, and activity of the biomolecule [1,10,21].

In the following sections are described the nanobioconjugates assembly and characterization based on the use of different nanomaterials such as noble metals, oxide metals, polymers, carbon allotropes, quantum dots (QDs), and biomolecules that include proteins, antibodies, aptamers, carbohydrates, and single and double DNA/RNA strands, among others. Hereafter, the review illustrates the use of these nanobioconjugates for signal amplification of biorecognition events in a variety of biosensor formats focused on molecular-, enzymatic- and electroactive complex-based strategies. Finally, it points out the current challenges and future perspectives of nanobioconjugates for signal amplification in biosensing.

## 2. Nanobioconjugates Assembly

The development of nanobioconjugates involves the coupling of different types of nanomaterials (or nanomaterial precursors) with a variety of biomolecules (Figure 1B). Biomolecules such as proteins, peptides, nucleic acids (i.e., DNA/RNA, oligomers, aptamers, and ribozymes/DNAzymes), lipids, fatty acids, and carbohydrates link to the nanomaterials for developing plenty of nanobioconjugate configurations, depending on the aim or the application.

### 2.1. Noble Metal Nanobioconjugates

Noble metals are characterized by their ability to resist corrosion and oxidation under exposure to oxidative conditions. Rhenium, ruthenium, rhodium, palladium, silver, osmium, indium, platinum, and gold are a select group of metals that are considered in the noble metal family [10].

Nanomaterials of noble metals synthesized in the form of nanoparticles, nanorods, nanoshells, nanocages, and hollow nanospheres, have been bioconjugated for the development of electrochemical biosensing platforms [7]. For example, Wahyuni et al. developed nanobioconjugates based on DNA and AuNPs to detect Sus scrofa mitochondrial DNA. The nanobioconjugation was achieved by attaching DNA probes onto an AuNPs surface, electroplated at an SPCE surface, through the 3-mercaptopropionic acid (MPA) linker by the EDC/NHS coupling chemistry. The DNA from the nanobioconjugates hybridized with the complementary target sequence (Sus scrofa mitochondrial) to form a dsDNA. Methylene blue (MB) that interacts with guanine bases on ssDNA and intercalates with dsDNA was used as a signal indicative of the target detection. In the presence of mitochondrial Sus scrofa DNA, the dsDNA complex gave an amplified signal response with respect to the ssDNA one, thanks to the higher quantity of MB molecules that were intercalated in between the dsDNA. This genosensor was interrogated by differential pulse voltammetry (DPV) at a potential window from −0.45 V to +0.1 V, reaching a linear range from 0.1 to 5.0 μg/mL, with a limit of detection (LOD) of 0.58 μg/mL [24]. 

Shamsipur et al. reported the development of a biosensor based on aptamers (Apt) bioconjugated with a silver nanocluster (AgNCs) for the impedimetric detection of cytochrome c (Cyt c), a small hemeprotein that stimulates the apoptotic process into the cytoplasm. The Apt@AgNCs were employed for the detection of apoptosis based on the Cyt c release from the mitochondria in lysates from human embryonic kidney cells. Ag(I) ions from an AgNO_3_ solution were employed for the nanobioconjugates synthesis, due to their high affinity to bind with oligonucleotide bases from the DNA aptamer. The as-developed Apt@AgNCs, chemisorbed onto a gold electrode (GE), served for the specific binding of Cyt c. The biosensor responded to changes in Cyt c concentrations in a linear range from 0.15 to 375 nM, with a LOD of 72 pM [25], when interrogated by electrochemical impedance spectroscopy (EIS).

Zhao et al. designed a biosensor that used a platinum and palladium nanocage (Pt@PdNCs) hybrid material to develop two DNA-based nanobioconjugates for the sensitive detection of Pb^2+^, as shown in Figure 2I. Bimetallic nanocages were first prepared with K_2_PdCl_4_ and H_2_PtCl_6_ as precursors of Pd and Pt, respectively. After such, the nanobioconjugation involved anchorage of DNA aminated sequences (S3 and S4 probes) and the adsorption of the electroactive toluidine blue. The biosensor assembly started with chemisorption of an aminated DNA hairpin (H1) on top of a glassy carbon electrode (GCE) modified with electroplated AuNPs. DNAzyme substrate strand (S1) and DNAzyme catalytic strand (S2) probes were then hybridized, forming a specific DNAzyme that was cleavaged in the presence of Pb^2+^. The resultant cleavage sequences (rS1) opened the H1 sequence, which was then displaced by the presence of a second hairpin H2 for n cycles. Then, the opened H2 probe hybridized first with the S3-Pt@PdNCs nanobioconjugate and this hybridized then with the S4-Pt@PdNCs one to form a network. Finally, (4-*N*-methylpyridiniumyl)-porphyrin (MnTMPyP) intercalated in between the dsDNA served as an enzyme mimic for the final signal response. The biosensor response was interrogated by DPV, being linear in range from 0.1 pM to 200 nM, with a LOD of 0.033 pM [26].

Tao et al. assembled an ultrasensitive electrochemical biosensor based on AuNPs conjugated with doxorubicin (AuNPs@Dox) for the detection of microRNA, a small endogenous molecule involved in the regulation of gene expression in the life process. First, the AuNPs@Dox@S1 nanobioconjugates were synthesized by using citrate stabilized AuNPs densely functionalized with Dox and thiolated DNA probes (S1). A thiolated capture probe was immobilized onto a gold pretreated electrode surface modified with mercapto-1-hexanol (MCH). Then, the AuNPs@Dox@S1 nanobioconjugate was dropped onto the electrode surface to allow the hybridization between the capture probe and S1 from nanobioconjugates. The S1 from the nanobioconjugates, hybridized with the capture probe, was displaced by the addition of different concentrations of miRNA let-7 target. The reported biosensor response, measured by EIS, was linear from 1 pM to 10 nM, with a LOD of 0.17 pM. The AuNPs@Dox nanobioconjugates in the biosensing platform allowed the ultrasensitive electrochemical detection and signaling amplification of miRNA, offering advantages in terms of sensitivity as compared to the northern blot and microarray assays, i.e., cost-affordable equipment and a straightforward operation system as compared to real-time PCR (RT-PCR) [27].

Some nanobioconjugates have been assembled into electrochemical platforms to detect the human breast carcinoma Michigan Cancer Foundation-7 cells (MCF-7). For example, Yazdanparast et al. designed an aptamer electrochemical biosensor based on the use of silver nanoparticles biofunctionalized with mucin-1 (Muc-1) aptamer (Apt-AgNPs). Muc-1 was covalently linked to the carboxyl groups of glutathione-coated AgNPs by the carbodiimide coupling chemistry. The Muc-1 was also covalently immobilized at a GCE covered with poly(glutamic acid)-modified multi-walled carbon nanotubes (MWCNT). MUC-1 from both the nanobioconjugates and the electrode surface specifically recognized MCF-7 cells, which were quantified by differential pulse anodic stripping voltammetry (DPASV). At optimized conditions, the biosensor was tested with different cell concentrations, reaching a linear working range from 1.0 × 10^2^ to 1.0 × 10^7^ cells/mL, with a LOD of 25 cells/mL [28]. 

For the same purpose, Wang et al. developed a cytosensor based on the use of polyhedral AuPd alloy nanoparticles (PH-AuPd NPs) conjugated with a DNA aptamer as a signal tag and a three-dimensional reduced graphene oxide (3D-rGO) as the transducer platform. The PH-AuPd NPs were prepared by using HAuCl_4_ and H_2_PdCl_4_ as Au and Pd precursors and ascorbic acid (AA) as a reducing agent in the presence of copper-(II), acetate (Cu(CH_2_COO)_2_) and octadecyl trimethyl ammonium chloride (OTAC). PH-AuPd NPs were then functionalized with streptavidin followed by immobilization of a biotinylated DNA aptamer (H2) (H2/SA-PH-AuPd). The transducer was a GCE modified with 3D-rGO and AuNPs to immobilize a thiolated aptamer H1 that interacted specifically with the MCF-7 cells. Thus, MCF-7 cells were sandwiched in between the aptamer-modified transducer and the nanobioconjugate to complete the biosensing assay, see Figure 2II. At optimal conditions, the cytosensors was tested by DPV with different target concentrations responding in a linear range from 50 to 10^7^ cells/mL, with a LOD of 20 cells/mL [29].

Coming back to the detection of cancer biomarkers, Zhou et al. proposed an electrochemical immunosensor based on the use of the hydroxyl pillar[5]arene@AuNPs@g-C_3_N_4_ hybrid nanomaterial bioconjugated with a secondary prostate-specific antigen (PSA) antibody (Ab2). The hybrid nanomaterial was synthesized by mixing a solution containing hydroxyl pillar[5]arene (HP5) with HAuCl_4_ and further functionalized with graphitic carbon nitride (C_3_N_4_) mediated by the pyrolysis of thiourea. Then, the hybrid nanomaterial was mixed with a solution containing MB and further incubated in an Ab2 solution to obtain the final HP5@AuNPs@g-C_3_N_4_-Ab2 nanobioconjugate. Otherwise, a primary antibody (Ab1) was adsorbed on top of AuNPs electroplated at the surface of a GCE. A solution containing the PSA was added to the electrode surface, followed by the nanobioconjugates. At optimal conditions, the immunosensor was tested by DPV with different PSA target concentrations, having a linear response in the range from 0.0005 to 10.00 ng/mL, with a LOD of 0.12 pg/mL, see Figure 2III [30].

Similarly, nanobioconjugates based on DNA sequences have been assembled to detect other cancer biomarkers. For instance, Mo et al. reported a DNA rolling photoelectrochemical biosensor for detecting oral cancer overexpressed 1 biomarker (ORAOV1) based on AuNPs bioconjugated with DNA sequences, for DNA rolling engineering. A thiolated DNA probe H1 was chemisorbed at the AuNPs surface (AuNPs-H1). The resultant nanobioconjugates started the catalytic hairpin assembly by using the DNA ORAOV1 target that opened the hairpin H1, and then, the target was displaced by the presence of the hairpin H2 getting the AuNPs-H1/H2 nanobioconjugate, which cycle was repeated several times. For the biosensor assembly, GO-modified GCE was decorated with hemin through π-π stacking interactions and PtNPs further electroplated to link DNA capture probes, which hybridized a fraction of H2 from the AuNP-H1/H2nanobioconjugate. Next, T7 exonuclease removed the nanobioconjugate-capture probe duplex, leaving a small fraction of the capture probe. The remaining fraction of the capture probe was incubated with an AgNO_3_ solution in which Ag^+^ ions interacted with high affinity with cytosine bases. Finally, the Ag^+^ ions were electrochemically reduced to produce silver nanoclusters (NCs) that served as a signal amplification tag, as shown in Figure 2IV. At optimized conditions, the developed biosensing platform was tested with different ORAOV1 concentrations. The platform was photo-stimulated to give a concentration-dependent photocurrent signal intensity in a linear range from 1 fM to 10 nM, with a LOD of 0.33 fM. The biosensor demonstrated the ability to test ORAOV1 in saliva samples with high sensitivity [31].

In the context of cancer biomarkers detection, Miao et al. employed DNA nanobioconjugates for the detection of microRNA-21. The AuNPs-ssDNA nanobioconjugates were synthesized by chemisorbing a thiolated ssDNA detection probe functionalized with MB on top of AuNPs that served as signaling tag. The biosensor was assembled at a thiolated capture probe-modified GE as a transducer platform. The miRNA-21 target hybridized in between the capture and the detection probe. A solution containing K^+^ ions and iridium (III) complex was added to the sandwich-type genosensor that led to the formation of G-quadruplex that catalyzed the reduction of hydrogen peroxide (H_2_O_2_) with MB as a mediator. Such electrochemical reduction was followed by cyclic voltammetry (CV), having a linear response in the range from 5.0 fM to 1.0 pM, with a LOD of 1.6 fM. The genosensor was tested in human serum samples demonstrating its high potential for the bioanalysis of DNA targets [32].

Nucleic acids have been combined with antibodies for the development of nanobioconjugates for the specific detection of antigens in multiple applications. Alizadeh et al. illustrated an electrochemical immunosensor based on AuNPs bioconjugated with ssDNA probes and anti-hepatitis B antibodies (HBsAg), for the specific detection of hepatitis B virus antigen. AuNPs were first biofunctionalized with anti-HBsAg antibody (Anti-HBsAg-AuNPs) and then treated with a solution of thiolated bar-code G-quadruplex ssDNA probe to produce the nanobioconjugate (Anti-HBsAg-AuNPs-ssDNA). It was then treated with a solution containing hemin and MB to form a DNAzyme based on the nanobioconjugate, where the hemin and MB interacted with ssDNA (Anti-HBsAg-AuNPs-hemin/G-quadruplex). The DNAzyme, in the presence of H_2_O_2_, acted as a redox pair. Another nanobioconjugate was prepared with a primary antibody covalently immobilized on top of iron oxide NPs (Fe_3_O_4_-Ab1). The immunocomplex was confined on top of a gold disk by a magnetic field and two solutions containing the antigen HBsAg and the nanobioconjugates, respectively, were added to the gold disk electrode surface to complete the biosensing assay. The immunosensor was tested by square wave voltammetry (SWV) with different HBsAg target concentrations, showing a linear response in the range from 0.3 to 1000 pg/mL, with a LOD of 0.19 pg/mL [33].

By using antibody-based nanobioconjugates, Khristunova et al. developed an electrochemical immunosensor based on AgNPs biofunctionalized with an antibody for the detection of tick-borne encephalitis antibody (anti-TBEV), which binds specifically with the antigen tick-borne encephalitis (TBEV), an endemic flavivirus that causes encephalitis/meningoencephalitis. AgNPs were prepared by the reduction of AgNO_3_ with NaBH_4_ and then mixed with a solution containing the anti-TBEV antibody to obtain the AgNPs@anti-TBEV nanobioconjugates. The transducer platform consisted of a gold-carbon composite electrode functionalized with cysteamine followed by glutaric aldehyde for the covalent linking of the TBEV antigen. The anti-TBEV was added to the electrode surface for the specific binding with the antigen, followed by the AgNPs@anti-TBEV nanobioconjugates to complete the biosensor platform. At optimal conditions, different antibodies concentrations were tested by cathodic linear sweep voltammetry (CLSV) by following the silver chloride reduction, which response was linear in the range from 50 to 1600 IU/mL, with a LOD of 50 IU/mL [34].

Zouari et al. reported a genosensor based on AuNPs bioconjugated with streptavidin and ferrocene (Fc) for the ultrasensitive detection of circulating microRNA in serum from cancer patients. Streptavidin was adsorbed on top of the AuNPs and further treated with a solution containing 6-ferrocenylhexanethiol to get the Fc-AuNPs-streptavidin nanobioconjugate. Afterward, rGO was electroplated at the surface of an SPCE and further coated with AuNPs (AuNPs/rGO/SPCE) to link a thiolated capture probe. The miRNA target sequence hybridized in between the capture probe and a biotinylated signal probe that was then coupled to the Fc-AuNPs-streptavidin nanobioconjugate. The as-developed biosensor was interrogated by DPV with different target concentrations, reaching a linear response ranging from 10 fM and 2 pM, with a LOD of 5 pM [35].

It is important to highlight that most of the nanobioconjugates based on noble metals described in this section were assembled with AuNPs bioconjugated with different biomolecules depending on the final purpose or biosensor use. The easiness of gold nanomaterials for functionalization by forming SAMs at their surface made them ideal materials for biomolecules conjugation. The AuNPs have demonstrated to be a proper ‘nanoenvironmet’ that promotes the controlled linking of bioreceptors while keeping its conformation, nature and activity. Moreover, their conductive properties permit the enhancement of the electrochemical performance when assembled in biosensing platforms. They are of particular usefulness in the ultrasensitive detection of target molecules not only associated with different pathologies, which are usually present at extremely low concentrations in a biological matrix but in the detection of traces in environmental applications, among many other applications.

Along with the signal amplification achieved by the incorporation of nanobioconjugates at biosensors, the choice of the proper electrochemical technique to interrogate them is also crucial to achieving high sensitivity. NPs or NSs of conducting materials, metals or QDs can be used in connection to a proper electrochemical technique for enhancing the signal response of a biorecognition event. For instance, there are many reports in the literature where the target analyte is followed by metallic NP- or NS-based nanobioconjugate dissolution, which signal response is registered by anodic (or cathodic) stripping voltammetry or anodic (or cathodic) redissolution voltammetry [36,37]. Figure 2 shows representative examples of electrochemical biosensors developed with noble nanomaterial-based nanobioconjugates and Table 1 summarizes the analytical features of the biosensors described.

### 2.2. Quantum Dot-Based Nanobioconjugates

QDs are known as inorganic semiconductor crystals with nanoscale size between 2 to 10 nm in diameter. QDs final application is a parameter highly dependent on the nature of the capping agent as a result of the synthetic procedure. The synthesis method leads to having QDs with different functional groups at their surfaces that allow different biomolecules’ immobilization [38]. QDs exhibit different properties as compared with noble metals, which are related to their material composition and dimensionality. The QDs behavior is different than the concomitant bulk particle counterparts. For example, QDs exhibit quantum confinement effects that are associated with the emission quantum yield, tunable emission profile and narrow spectral band that gives them different optical properties [39] and semiconductor behavior [40]. The optical properties of QD nanobioconjugates offer great opportunities for the development of biosensing strategies [7,39].

Li et al. developed an immunosensor based on cadmium selenide (CdSe)-QD-melamine networks for the detection of the carcinoembryonic antigen (CEA) cancer biomarker, see Figure 3I. The CdSe-QDs and the amino groups of each melamine molecule form a strong network to link later a secondary antibody anti-CEA (Ab2) through the EDC/NHS coupling chemistry (CdSe-Ab2). On the other hand, an indium-tin-oxide (ITO) was coated first with TiO_2_ by ultrasonication and then with dispersed AuNPs to obtain an Au-TiO_2_ hybrid-ITO transducer platform, where a primary anti-CEA antibody (Ab1) was linked. A solution containing CEA was added so that the CEA was sandwiched in between the Ab1 and Ab2. After photo-stimulation, the immunosensor was challenged with different antigen concentrations producing different photocurrent signal responses. At optimal conditions, it responded linearly in a range from 0.005 to 1000 ng/mL, with a LOD of 5 pg/mL. Additionally, the as-developed biosensor showed to have high stability and reproducibility and high performance when analyzing real serum samples [41].

Yang et al. designed an ultrasensitive electrochemical biosensor based on QDs bioconjugated with DNA probes for the specific detection of microRNA of breast cancer 1 gene mutation (BRCA1), as shown in Figure 3II. The nanobioconjugates were prepared by functionalization of the cadmium telluride (CdTe) QDs with mercaptoacetic acid followed by the immobilization of amino ssDNA1 and ssDNA2 probes, thus forming a CdTeQD@DNA 3D-structure. The transducer used a GE functionalized with a thiolated hairpin capture probe. The BRCA1target sequence was dropped onto the electrode surface and opened the hairpin capture probe as a result of the hybridization process. Then, a double-strand specific nuclease (DSN) was added to cleave the target-capture ds-DNA probe, thus leaving a short fragment of the DNA probe at the electrode. Finally, the 3DCdTeQD@DNA nanobioconjugate was added to the electrode surface to hybridize with the capture probe fragment. The as-developed biosensor was tested by DPV with different target BRCA1 concentrations, showing a linear working range from 5 aM to 5 fM, with a LOD of 1.2 aM [42].

By exploring the use of QDs for signal amplification, Liu et al. proposed nanobioconjugates based on concatemer-QDs linked to a DNA-aptamer for the ultrasensitive detection of cancer cells. CdTe QDs coated with carboxyl acid groups were functionalized with an aminated DNA-aptamer by the EDC/NHS coupling chemistry. The concatemer-QD/DNA-aptamer was prepared by mixing the DNA-aptamer/QD nanobioconjugates with signal DNA sequences (SDNA) that hybridized not only with the DNA-aptamer sequences but with the target DNA, being specific for the cell biorecognition. The as-developed concatemer nanobioconjugate was employed as a signal amplification tag in the cytosensors development with a T lymphoblastoid line obtained from the peripheral blood (CCRF-CEM) as a model of cancer cells. The developed cytosensor tested by anodic stripping voltammetry (ASV), responded with different target concentrations in a linear working range from 10^2^ to 10^6^ cells/mL, with a LOD of 50 cells/mL [43].

Qin Yan. et al. proposed an electrochemical label-free immunosensor based on graphene quantum dots (GQDs) supported on a hybrid AuPdCu nanoparticle for the specific detection of hepatitis B antigen (HBsAg). For this purpose, CuCl_2_, HAuCl_4_, and PdCl_2_ were the precursors of Cu, Au, Pd, respectively, for the preparation of AuPdCu nanoparticles. GQDs, prepared from citric acid and dicyandiamide by the hydrothermal method, were incorporated into the AuPdCu nanoparticles by a hydrothermal treatment to produce AuPdCu/GQDs. Then, such a hybrid nanomaterial was immobilized on top of polymer nanospheres (PS) by electrostatic interactions to render the AuPdCu/GQDs@PS nanocomposite, which was employed to coat the working electrode of a GCE and immobilize the hepatitis B antibody (anti-HB) on top. The immunosensing platform performance, interrogated by EIS, was HBsAg concentration-dependent in a linear range from 10 fg/mL to 50 ng/mL, with a LOD of 3.3 fg/mL [44].

Tufa et al. assembled an electrochemical immunosensing platform for the specific detection of *Mycobacterium tuberculosis* antigen, (CFP-10). The immunosensor used core-shell nanostructures based on a Fe_3_O_4_@Au composite coated with GQDs to immobilize a primary antibody at the electrode surface and nanobioconjugates based on AuNPs functionalized with a secondary antibody (Ab_2_) as a signaling tag. The core-shell nanostructures were developed by using Fe_3_O_4_ coated with poly(ethyleneimine) (PEI) that serves for the in-situ functionalization of AuNPs. The Fe_3_O_4_@Au composite was decorated with a silver layer followed by modification with PEI to allow the immobilization of GQDs and produce the Fe_3_O_4_@Au@Ag/GQDs nanoparticles. The resultant Fe_3_O_4_@Au@Ag/GQDs were attracted to a GCE to immobilize an anti-CFP-10 primary antibody and bind specifically the CFP-10 antigen, while the AuNPs-Ab_2_ nanobioconjugate was added to the electrode to complete the biosensing platform. The immunosensor signal response was interrogated by DPV, reaching a linear range from 0.005 to 500 µg/mL, with a LOD of 0.33 ng/mL [45].

Rezaei et al. produced an enzyme-free electrochemical biosensor for the simultaneous detection of two biomarkers of hemophilia A, which involved two methodologies for signal amplification. One was based on a target recycling molecular amplification process and the other one on QDs encapsulated into metal-organic frameworks, functionalized with S1 and S2 DNA probes. The encapsulated QDs were prepared from PbCl_2_, CdCl_2_ and mercaptoacetic acid (MPA) followed by the addition of sodium sulfide to produce MPA-PbSQDs and MPA-CdSQDs. After that, both hybrids were treated with 2-methylimidazole to encapsulate the QDs into a zeolitic imidazolate framework-8 (ZIF-8) to form QDs@ZIF-8. Then, PbS and CdS QDs@ZIF-8 were decorated with PEI followed by the immobilization of S1 and S2 DNA probes, respectively. Two hairpin probes H1 and H2 were chemisorbed at a gold-modified GCE and T1 (miR-1246) and T2 (miR-4521) DNA target probes were dropped onto the electrode surface to hybridize with the hairpin H1 and H2, respectively. Then, the PbS@ZIF-8-S1 and CdS@ZIF-8-S2 nanobioconjugates were added to the electrode to allow the displacement of the target sequence to open the H1 and H2 hairpin probes for n times. Finally, a solution containing HCl was employed to dissolve the encapsulated PbS and CdS QDs to measure the released Pb(II) and Cd(II) ions by DPV. The biosensor response was linear from 1 fM to 1 µM, with a LOD of 0.19 and 0.28 fM for miR-1246 and miR-4521, respectively [46].

### 2.3. Carbon Nanobioconjugates

Single and multi-walled carbon nanotubes (CNTs), graphene and its oxide (GO) are the most studied allotropes of carbon. Structural, electrical, mechanical, and optical properties of these carbon nanostructures made them fascinating candidates for application in the biosensing field [8,47]. For example, a high surface-to-volume of carbon allotropes is ideal when implemented in biosensors [48,49], not only as a part of the transducer platform [50], but as a labeling tag [51] for signal amplification.

Farzin L et al. developed an electrochemical aptasensor based on nanobioconjugates assembled by coupling an NH_2_-termini aptamer to reduced graphene oxide (rGO), which served a signaling tag in a sandwich-like biosensor for the specific detection of mucin 1 (MUC1). rGO was prepared by the hummer’s method, followed by functionalization with malononitrile anion (rGO-CN). Hydroxylamine hydrochloride was added to the rGO-CN to reduce the CN groups from the rGO surface to obtain rGO-*N*′1,*N*′3-dihydroxymalonimidamide and then a solution containing thionine (Th) was added to form rGO/Th. The aptamer anti-MUC1 was immobilized at the functional rGO/Th platform by crosslinking with glutaraldehyde (anti-MUC1-rGO/Th) and the biosensor was assembled at a GE coated with a hybrid compound of Nafion/polymer dots. The MUC1 aptamer was linked to the electrode surface by covalently coupling the amino groups from the MUC1 and the carboxyl groups from the polymer dots by the EDC/NHS method. A solution containing MUC1 was added to the electrode surface to happen the aptamer-protein binding. Finally, the anti-MUC1-rGO/Th nanobioconjugates were added to the electrode surface to complete the biosensing platform. At optimal conditions, the biosensor response, interrogated by DPV, was target concentration-dependent in a linear range from 0.1 to 14.5 nM, with a LOD of 0.06 nM [52].

CNTs have been the platform of choice for the development of a variety of immunosensors. For example, Palomar et al. reported the design of an impedimetric immunosensor for specific detection of anti-cholera toxin antibody. The immunosensor was developed by using a GCE modified with MWCNTs. The modified electrode served for the immobilization of poly-pyrrole-nitriloacetic acid (poly-NTA) by chronopotentiometry, followed by the addition of Cu^2+^ ions to form a complex with NTA molecules. A solution containing the target cholera toxin was added to the electrode to bind with Cu^2+^ by coordination bonds. Next, a solution containing the anti-cholera antibody was dropped onto the electrode surface to complete the biosensing platform. Finally, a solution containing the Fe(CN)_6_^3−^ redox probe was added to the electrode to investigate the biosensor performance by EIS. The as-developed biosensor reached a linear-dependent concentration response in the range from 0.1 pg/mL to 0.1 µg/mL, with a LOD of 0.1 pg/mL [53].

Sánchez-Tirado E. et al. developed an immunosensor for the determination of transforming growth factor β1 (TGF-β1) cytokine based on CNT bioconjugates, as shown in Figure 3III. First, the nanobioconjugates were prepared by using single-walled carbon nanotubes (SWCNTs), which were modified with 4-aminobenzoic acid to obtain carboxyl groups onto the SWCNTs surface. Then, the SWCNTs-COOH were treated with (3-aminoethyl)-4,4′-bipyridinium bromine to get the viologen-SWCNTs (v-SWCNTs), where the anti-TGF-β1 antibody was immobilized. Immobilization was achieved by using terephthalic acid that acts as a linker between amino groups from the hybrid material and those from the antibodies. Next, a solution containing HRP was added to the anti-TGF-β1-v-SWCNT nanobioconjugate to obtain the anti-TGF-β1-v-SWCNT/HRP complex. In parallel, the biosensor was constructed onto streptavidin-coated SPCE to link a biotinylated anti-TGF-β1. Finally, a solution containing the antigen TGF-β1 was dropped onto the electrode surface, followed by the addition of the anti-TGF-β1-v-SWCNT/HRP nanobioconjugate, and the signal response was interrogated by amperometry. The results showed a target-dependent concentration linear in the range from 2.5 to 1000 pg/mL, with a LOD of 0.95 pg/mL [54].

Other examples include the immunosensor proposed by Liu Y et al. for the detection of squamous cell carcinoma antigen (SCCA) based on two different nanobioconjugates, one to be implemented as transducer platform and the other one as labeling tag, respectively. The nanobioconjugates were prepared by using GO functionalized either with β-cyclodextrin (β-CD) (CD-GO) or with Pd, Pt, and Cu (Pt/Pd/Cu-GO). Pt/Pd/Cu-GO was functionalized with a secondary antibody Ab2 to produce Ab2-Pt/Pd/Cu-GO. The transducer was a GCE decorated with CD-GO that allowed the immobilization of a primary antibody Ab1. A solution containing the antigen SCCA was added to the electrode surface, followed by the nanobioconjugate Ab2-Pt/Pd/Cu-GO, as shown in Figure 3IV. The immunosensor performance was tested by EIS, showing linear ranges from 0.0001 to 1 ng/mL and 1–30 ng/mL, with a LOD of 25 fg/mL [55].

Li et al. built an electrochemical immunosensor for the detection of CEA. A nanocomposite was prepared by decorating some rGO flakes with pyrenebutyric acid (PY) to introduce carboxyl groups to the material by π-π stacking interactions (rGO-PY) to link aminated polyethylene glycol (PEG). A GCE was coated with the as-prepared nanocomposite by ultrasonication, followed by electroplating AuNPs. Finally, an anti-CEA antibody was immobilized onto the electroplated AuNPs to capture the antigen CEA to complete the platform. The biosensor performance was interrogated by EIS, showing changes in the impedance in a CEA concentration-dependent manner in a linear range from 0.1 to 1000 ng/mL, with a LOD of 0.06 ng/mL [56].

### 2.4. Magnetic and Metal Oxide Nanobioconjugates

Magnetic nanoparticles (MNPs) are an important type of nanomaterials, which properties may be tailored based on their dimensions, material composition, structures and synthetic routes that they are produced [57]. Functional magnetic nanomaterials have gained increasing attention as a result of their unique physical properties, especially their high surface area and easiness of separation under external magnetic fields [58]. Their applications include the purification of biomolecules, cell separation, immunoassays, magnetic resonance imaging, drug delivery, and biosensing [59,60]. Particularly in the biosensing field, many authors have published magnetic and metal oxide nanomaterials to produce biomolecule-based nanobioconjugates.

By way of illustration, Wang et al. designed a nanobioconjugate-based immunosensor for the detection of amyloid-β protein, a peptide involved in Alzheimer’s disease due to its extremely strong neurotoxicity, see Figure 3V. The immunosensor used nanobioconjugates for both the transducer platform and the signal amplification. The signaling tag was prepared from a cerium oxide-zinc oxide nanoflower (Ce: ZONF) hybrid nanomaterial synthesized by a hydrothermal method, with lysine as a reducing agent that also generated carboxylated groups at the surface to link glucose oxidase (GOD) luminol and anti-Aβ (Ab2) by covalent coupling to produce Ab2-GOD@Ce:ZONFs-Lum. The immunosensor was assembled at an AuNP-modified GCE and then silver nanowires were attached to the modified Au-GCE to serve as the binding point for an Ab1 antibody. The immunosensor responded to changes in amyloid-β protein antigen concentrations, whose performance was evaluated by the intensity of electrogenerated chemiluminescence (ECL), showing a linear range from 80 fg/mL to 100 ng/mL, with a LOD of 52 fg/mL [61].

MNPs are appealing for biosensing development as a result of their unique magnetic properties, as mentioned. These properties make easy supernatant removal, sample preconcentration, and purification [62]. In this context, they are ideal candidates for the development of nanobioconjugate- based biosensors for the detection of cancer biomarkers. To show one example, Yu et al. reported an ultrasensitive electrochemical immunosensor based on an antibody (Ab2) bioconjugated on top of an Au@CeO_2_ hybrid nanomaterial for the detection of tumor-specific growth factor (TSGF). The hybrid nanomaterial was synthesized by a hydrothermal method and further functionalized with a secondary antibody, serving as a label tag in the immunosensor assembly. The transducer was prepared by grafting reduced graphene oxide-tetraethylene pentaamine (rGO-TEPA) at a GCE and coupling a primary anti-TSFG antibody (Ab1) by crosslinking with glutaraldehyde. The Ab2-Au@CeO_2_ nanobioconjugate-based immunosensing platform responded in a TSFG concentration-dependent manner when tested by EIS, in a linear range from 0.5 to 100 pg/mL, with a LOD of 0.2 pg/mL. Furthermore, the as-developed immunosensing platform holds the potential for implementation in a variety of biosensing formats to test cancer biomarkers [63].

For the detection of the CEA cancer biomarker, Feng et al. developed an immunosensor with ferroferric oxide@silica–amino groups (Fe_3_O_4_@SiO_2_–NH_2_) bioconjugated with a secondary antibody (Ab2) as tags. Fe_3_O_4_ MNPs were first synthesized by a solvothermal process and then silanized with tetraethyl orthosilicate (TEOS) and (3-aminopropyl) triethoxysilane (APTES) (Fe_3_O_4_@SiO_2_–NH_2_) to have MNPs coated with amino groups onto their surface. Then, ferrocene (Fc) was immobilized at the MNPs and an Ab2 was linked to the surface by crosslinking with glutaraldehyde. The transducer platform was a GO-AuNP-grafted GCE, where the primary antibody, anti-CEA Ab1, was immobilized. The CEA antigen was then in a sandwich in between Ab1 and Ab2, in which concentration was interrogated by DPV. The nanobioconjugate responded to different target concentrations and amplified the resultant signal in a dynamic linear range from 0.001 to 80 ng/mL, with a LOD of 0.0002 ng/mL [64].

With the same purpose, the immunosensor reported by Peng et al. was based on Fe_3_O_4_ NPs functionalized with AuNPs bioconjugated with a secondary antibody (Ab2), i.e., (Fe_3_O_4_/AuNPs/Ab2). The Fe_3_O_4_ were synthesized by a solvothermal method and then treated with APTES to attach the AuNPs and further adsorb a secondary antibody (Ab2). The transducer platform consisted of a GCE decorated with GO and chitosan previously functionalized with ferrocene. The primary antibody (Ab1) was linked to the transducer by the EDC/NHS covalent coupling reaction, which recognized specifically the antigen CEA in a concentration-dependent manner when sandwiched with Fe_3_O_4_/AuNPs/Ab2 nanobioconjugates. The signal response was triggered by the ferrocenium ion that oxidizes p-aminophenol (AP) to produce p-quinone imine, which is then reduced to AP by NaBH_4_. At optimal performance conditions, the immunosensor detected different CEA target concentrations by CV, in a linear range from 0.001 to 30 ng/mL, with a LOD of 0.39 pg/mL [65].

As another illustration of MNPs for cancer detection, Shamsipura et al. described an immunosensor based on Fe_3_O_4_-secondary antibody (Ab2) bioconjugates for the specific detection of human epidermal growth factor receptor 2 (HER2), an important biomarker of oncogenic treatment for breast cancer. The Fe_3_O_4_ NPs were synthesized by coprecipitation of the FeCl_3_ and FeCl_2_ precursors. They were further functionalized with 3-aminopropyltrimethoxysilane (APTMS) to link the anti-HER2 antibody by crosslinking with glutaraldehyde (Fe_3_O_4_-APTMS-anti-HER2). Another nanobioconjugate was prepared by using Fe_3_O_4_-APTMS functionalized with AuNPs to attach a thiolated primary anti-HER2 antibody followed by immobilization of hydrazine (hyd) to obtain the antiHER2/Hyd@AuNPs-APTMS-Fe_3_O_4_ nanobioconjugate. The transducer was built at a GCE coated with the Fe_3_O_4_-APTMS-anti-HER2 nanobioconjugate. A solution containing the antigen HER2 was then added to the electrode surface, followed by the addition of the antiHER2/Hyd@-APTMS-Fe_3_O_4_ nanobioconjugate. Finally, the electrode was immersed in a silver nitrate solution, where silver ions were reduced by hydrazine from the AuNPs surface. The biosensor response, registered by square wave stripping voltammetry (SWSV), was linear from 0.5 pg/mL to 50 ng/mL, with a LOD of 0.02 pg/mL [66].

Hartati et al. developed an immunosensor based on cerium oxide NPs bioconjugated with a secondary anti-HER2 (CeO_2_-anti-HER2) antibody for the specific detection of HER2. CeO_2_ NSs were synthesized by a hydrothermal method and further functionalized with APTES first and with polyethylene glycol-α-maleimide-ω-NHS (NHS-PEG-Mal) later to obtain the NHS-PEG-Mal-NS CeO_2_ hybrid material. The anti-HER2 antibody was thiolated with 2-iminithiolane and chemisorbed on top of the anti-HER2/NHS-PEG-Mal-NS CeO_2_ hybrid. The biosensor was assembled at AuNP-decorated SPCE functionalized with a SAM of mercaptopropionic acid to link cysteamine by the EDC/NHS method. Then, the anti-HER2/NHS-PEG-Mal-NS CeO_2_ nanobioconjugates were coupled to the electrode surface through the maleimide chemistry. After that, a solution containing the antigen HER2 was dropped onto the electrode surface for the specific antigen-antibody interaction. The immunosensor was tested by CV with different HER2 target concentrations, showing a linear response ranging from 0.001 to 0.5 ng/mL and 0.5 to 20 ng/mL, with a LOD of 34.9 pg/mL [67].

Dong et al. reported nanobioconjugates based on aptamer-DNA functionalized-MNPs implemented in a biosensor for tumor exosomes detection. Carboxyl acid groups at the outermost surface of 2–3 μm magnetic particles were covalently coupled with an amine-containing aptamer-DNA by the EDC/NHS chemistry. Such aptamer hybridized with 3 different multimessenger DNA strands forming nanobioconjugates with long DNA sequences. The formed nanobioconjugate interacted with the target liposome while releasing the three multimessenger DNA strands. The released DNA hybridized with a capture probe anchored onto a GE surface to form a double-strand DNA that, in the presence of a restriction enzyme, was fragmented, thus giving an electrochemical response measured by DPV, see Figure 3VI. The exosome’s biosensor response was linear in a range from 1000 to 120,000 particles/μL, with a LOD of 70 particles/μL [68]. Moreover, the methodology employed for the detection of exosomes in a complex biological system could be implemented in real sample diagnosis.

### 2.5. Polymeric Nanobioconjugates

Linear polymers are considered as one of the best molecular structures for the development of polymer bioconjugates. Their biological properties, synthetic practicality and easiness of functionalization make them of utility for biosensing applications [69]. Polymeric nanomaterials have gained increasing attention in the biosensing field because of the easiness of biomolecules’ functionalization and long-term stability of the functionalized polymers [70]. In the development of electrochemical biosensor, some polymers are also part of the transducer platforms, e.g., electroactive polymers that are easily electrodeposited onto the electrode surface, with controllable layer thickness and a variety of moieties for further surface functionalization [71].

Polymeric-based nanobioconjugates have also been used for cancer biomarker detection. In particular, Pan et al. developed an electrochemical biosensor based on carboxylated poly-lactide nanoparticles (PLLA) bioconjugated with a dual-antibody vascular endothelial growth factor antibody (anti-VEGF) and a PSA antibody (anti-PSA) for early diagnosis of prostate cancer. For this purpose, carboxylated PLLA in methylene chloride was treated with EDC and NHS to obtain PLLA-NHS that was precipitated with ethyl ether. PLLA-NHS was dissolved in chloroform and treated with PEG 2-aminoethyl ether acetic acid (NH2–PEG–COOH) and *N,N*-diisopropylethylamine, resulting in the PLLA-PEG-COOH block co-polymer. The nanoparticles were then formed by mixing the block co-polymer in dimethylformamide (DMF) with water by sonication. The resultant PLLA-PEG-COOH nanoparticles were treated with EDC/NHS to conjugate the anti-VEGF and anti-PSA antibodies. GO was modified with PEI by the EDC/NHS coupling chemistry and then treated with sulfosuccinimidyl4-(*N*-maleimidomethyl) cyclo-hexane-1-carboxylate (sulfo-SMCC) to immobilize a thiolated DNA (GO-ssDNA). A GE was then coated with the GO-ssDNA nanobioconjugate, whose ssDNA probes were specific for the VEGF antigen biorecognition. A solution containing the PLLA-PEG/anti-VEGF/anti-PSA nanobioconjugate was added to the electrode surface for the biorecognition of the VEFG antigen. Finally, the PSA antigen was also added to the electrode surface, which was recognized by the anti-PSA. The biosensor was tested by DPV, with different VEGF and PSA target concentrations, showing a linear response from 0.05 to 100 ng/mL and 1 to 100 ng/mL, with a LOD of 50 pg/mL and 1 ng/mL of VEGF and PSA, respectively. This reported methodology can be implemented for the rapid detection of two types of cancer biomarkers at the same time [72].

Table 2 summarizes electrochemical biosensors developed for the specific detection of different biomolecules that includes DNA, miRNA, antigens, proteins, cancer biomarkers and cells, most of them associated with human pathologies. Nanobioconjugates can be incorporated either to the transducer platform or to the signaling tag, always aiming to achieve an enhanced signal response as a result of the nanomaterial-biomolecules improved properties. In this context, an enhanced electrochemical response has been achieved by ultramicroelectrode arrays [73,74] and nanoelectrode arrangements [75]. The surface area is a critical aspect to be considered when assembling nanobioconjugates, i.e., the higher the surface area, the more biomolecules hosted and more amplified the signal response.

## 3. Characterization of Nanobioconjugates

There are many physicochemical and bioconjugation metrics that are of interest in the characterization of nanobioconjugates. They include purity, size, shape, particle or conjugate mass, aspect ratio, surface area, polidespersity and colloidal stability. Composition, surface properties, ζ potential and hydrodynamic radius, are other usual parameters to be considered when studying nanobioconjugates. Biomolecular orientation within the nanobioconjugate and activity, affinity, or avidity of the final conjugate for the target analyte is also necessary for interrogation [10]. Dynamic light scattering (DLS) [76], electrophoretic light scattering (ELS) microscopy [77], spectroscopy [77,78], and thermal [79] techniques are the most used for nanobioconjugates characterization and are described as follows.

Scattering techniques give quantitative information about the size, shape, charge, distribution and concentration based on the interaction of incident radiation with colloidal particles and nanobioconjugates. Among them, DLS is one of the most employed techniques for nanobioconjugates characterization. This technique gives information about the size and concentration of nanoparticles measuring the hydrodynamic particle size. Brownian motion allows for the estimation of their diffusion coefficient, which is directly correlated with the hydrodynamic radius by the Stoke–Einstein equation. Analysis of the sample is comparatively rapid, simple, cheap, non-invasive, and non-destructive but not that straight forward when the samples are polydisperse, making necessary microscopy techniques for more accurate characterization of such samples [1]. ELS is a measure of the ζ potential and gives information about the net superficial charge of nanoparticles or nanobioconjugates. The ζ potential is determined by applying an electric field to the sample where the velocity at which nanoparticles move toward an electrode of opposite charge is proportional to the ζ potential. It is indicative of the nanobioconjugates stability, which magnitude is correlated with the repulsion interactions and steric hindrance effects among adjacent charged particles in the colloidal suspension. The measurement of the ζ potential helps to determine if the bioconjugation process took place by taking into account the chemical nature of biomolecules [1].

X ray diffraction (XRD) is very useful to give information about the crystalline structure of the samples. This technique is powerful for the characterization of nanomaterials embedded inside biological matrixes or nanobioconjugates. The d-spacing analysis (the distance between crystallographic planes) is a parameter whose magnitude changes after a biomolecule is immobilized onto a nanomaterial and thus can be used to investigate the biomolecule orientation [1].

Microscopic techniques are based on the sample characterization by the use of light, electrons, and scanning probes. Scanning electron microscopy (SEM) gives information about surface topography and composition by the collection and processing of signals as a result of electrons striking the sample. SEM may have a resolution from 10 µm to 10 nm, but in some cases, the resolution can be down to 1 nm, depending on the equipment setup, operating parameters and sample material. The sample surface needs to be conducive to facilitate the microscopy analysis, usually achieved by depositing a conductive coating over the material before the SEM observation. Moreover, energy dispersive X-ray (EDX) can be employed under SEM analysis to determine the chemical composition of the sample. The released energy through photoemission in SEM-EDX depends on the electron configuration of the atoms and its collection allows the establishment of the elemental sample composition [80]. In transmission electron microscopy (TEM), a beam of electrons overpasses the sample to form an image. TEM may give information about the sample core, including biomolecules or nanobioconjugate-containing nanomaterials. The sample thickness must be less than 100 nm to reach the signal-to-noise ratio needed for high contrast, thanks to the very strong incident beam of electron-sample interactions. TEM provides information about morphology, crystallographic degree, crystallographic planes, nanomaterials defects, etc., based on analysis by diffraction, spectroscopic methods, and imaging. High-resolution scanning transmission electron microscopy (HR-STEM) may resolve at the atomic level, depending on the medium that supports the particles [81]. Atomic force microscopy (AFM) operates through a scanning probe and gives information about the topography, size and shape of the nanomaterials and biomolecules, as well as adhesion and other interactions in the nanobioconjugates [1]. Unlike SEM and TEM, AFM may image conductive and nonconductive samples in noncontact (static) and contact (dynamic) analysis by different probes commercially available.

Spectroscopic techniques take advantage of the electromagnetic radiation and its interaction with the samples that result in an absorption or emission spectrum that is directly dependent on wavelength and can be correlated with the size of nanobioconjugates and interactions among them. Ultraviolet and visible spectroscopy (UV-Vis) measures the interaction between electromagnetic waves and samples, giving information about the emitted or absorbed electromagnetic radiation by atomic or molecular species. The energy is supplied to the sample in the form of heat, light or chemicals. Each molecule absorbs the energy with a characteristic frequency and emits radiation, which intensity is a function of the wavelength. The spectroscopic analysis comprises atomic- and molecular-spectrochemical analysis and emission and absorption spectrum analysis. These techniques are used for rapid estimation of the size of the nanoparticles and nanobioconjugates by optical changes coming from collective oscillations and from characterizing chromogenic molecules or materials. UV-Vis spectroscopy is a fast, cheap, simple, non-destructive, and easy-to-operate technique [82].

Fourier transformed infrared spectroscopy (FT-IR) is the choice for rapid and easy characterization of functional groups of nanomaterials and nanobioconjugates. FT-IR radiation represents the molecular absorption and transmission as a result of the vibrational stretching and bending of molecules from the nanobioconjugates that create a fingerprint of a sample. FTIR is a non-destructive analysis technique where the intensity of peaks is directly correlated with the number of functional groups in the sample. It is a very useful technique for nanobioconjugates characterization in which the bioconjugation process is evaluated by comparing the spectrum before and after bioconjugation [83].

Electrochemical characterization (EC) consists of a set of powerful tools to evaluate and characterize the capacity of nanomaterials to be used mainly in energy storage and sensor applications. The EC is based on the evaluation of the mechanism involved in electron transfer, electron and mass transport and electrolyte behavior. EC techniques include cyclic voltammetry (CV), chronoamperometry (CA), chronopotentiometry (CP), galvanostatic charge-discharge (GCD), and electrochemical impedance spectroscopy (EIS), among others. EC studies the electrochemical performance of nanomaterials and nanobioconjugates under settled electrochemical conditions [84].

## 4. Nanobioconjugates in Biosensing

Biosensing refers to (bio)systems that can detect organisms, target analytes (biomolecules), and their biological activity [85,86] through electrical, thermal or optical signals, among other transduction mechanisms [87]. Biosensors are analytical devices that utilize a biological component in direct contact with a solid platform (transducer), which selectively respond to an analyte in a concentration-dependent manner. The resultant signal from the specific biomolecular interaction is read and registered in a simple way [88,89,90]. A biosensor must be designed to be versatile and easy to operate and have high-throughput, rapidity and accuracy [91]. Biosensors are labeled or label-free, depending on whether they use a mark or not to evidence a biorecognition event. Label-free biosensor refers to biosensing systems that only require one recognition element that reduces the assay time, reagent cost and the platform assembly but is restringed when the target concentration is too low.

In contrast, labeled biosensors can amplify the signal by incorporating nanobioconjugates as signaling tags [92]. In labeled-biosensor approaches, the target may be trapped in between a capture and a signal bioreceptor in a sandwich-like format. Whereas most of the capture bioreceptors are attached to nanostructured solid supports, electrodes, or chips, the signal bioreceptors are attached to some signaling tags, such as fluorophores, enzymes or nanomaterials. Signaling tags may have different binding sites, which enhance the signal and reduce the background [92,93], thus leading to an amplified response. Signal amplification takes advantage of the signaling tag, having more than one signal amplifier by bioreceptor in the same format. Signal amplification may also come from modified transducer platforms with electrodeposited materials or polyelectrolytes, which increase electron transfer in electrochemical biosensors and, in the presence of mediators, communicate with the nanobioconjugates-based signaling tags [94,95,96]. Researchers have explored different strategies to amplify the target signal and some of them are commented on as follows.

### 4.1. Molecular Strategies

Rolling circle amplification is an isothermal nucleic-acid methodology that, combined with nanobioconjugates, has been used as an amplification strategy in biosensor development. It requires a short target DNA, a circular template, deoxynucleotide triphosphates (dNTPs) and DNA polymerase [97,98,99]. Su et al. reported the use of RCA to detect and amplify the DNA of the hepatitis B virus, using QDs and a ruthenium complex as a reporter, acting as intercalant DNA and measured by fluorescence. The biosensor showed a linear range from 3.3 to 33 pM, with a LOD of 0.5 pM [98].

The catalyzed hairpin assembly (CHA) has been employed as a DNA amplification strategy. The principle of this methodology is the use of two hairpins (hairpin 1 and hairpin 2), which in the absence of target cannot hybridize each other. The target allows to open the hairpin 1; then, the hairpin 2 displaces the DNA strand because it is larger than the target. The released target can hybridize with another hairpin 1 and will be replaced by another hairpin 2 and so on. The hairpin 2 has a signaling tag that allows for recognition with a substrate that produces a signal, proportional to the target concentration [100,101]. Shuai et al. developed a biosensor for detecting microRNA based on (CHA) (see Figure 2). A tungsten and oxide graphene composite was used as a supporting platform, which was decorated with AuNPs to chemisorb the hairpin 1 on. The hairpin 2, having biotin as a signaling tag in its 5′ end, attached to streptavidin-conjugated alkaline phosphatase (ALP), see Figure 4I. This enzyme catalyzed the production of ascorbic acid (AA), thereby generating an electrochemical response in the presence of ferrocene-methanol by DPV, linear in the range from 0.1 fM to 100 pM, with a LOD of 0.05 fM [100].

The hybridization chain reaction is another strategy for signal amplification of DNA, also known as super-sandwich [102,103,104]. Based on this approach, the target can act as an initiator of the hybridization reaction and uses two hairpin sequences that hybridize one another to form an enlarged DNA duplex. As shown in Figure 4II, Shuai et al. reported the synthesis of a biosensor for detecting DNA sequences. This biosensor used two-dimensional tungsten disulfide–acetylene black (WS_2_–AB) and AuNPs dispersed onto the WS_2_-AB composite. The capture probe was immobilized on top of the AuNPs from the transducer platform by chemisorption and hybridized the target sequence, which in turn hybridized an auxiliary DNA strand. The auxiliary DNA hybridized with two hairpin sequences triggering a chain reaction, which incorporated multiple biotin anchoring points. Avidin-HRP (horseradish peroxidase) bond to the multiple biotin points acting as signaling tags, measured by DPV. The biosensor had a linear response between 0.001 to 100 pM and a LOD of 0.12 fM [105].

### 4.2. Enzymatic Strategies

Enzymes have been employed in biosensor development as signal amplifiers because of their high activity through redox reactions [106,107]. Shen Q. et al. used a methodology to detect and amplify the signal of human T-cell lymphotropic virus type II (HTLV-II). The target could be recognized using a nanocomposite based on graphene-cadmium sulfide and manganese NPs. A DNA probe was conjugated on the nanocomposite surface trough EDC-NHS coupling reactions between carboxyl groups of 3-mercaptopropionic acid and the NH_2_ groups of the DNA probe, which hybridized with the target probe. Multiplex biotinylated deoxyuridine triphosphate (dUTP) was added to the 3′ terminal target that allowed to prolong the DNA chain mediated by the presence of deoxynucleotidyl transferase (TdT), with multiple biotin points. The extended multiplex dUTP-biotin was able to bind with avidin-ALP, which catalyzed 2-phospho-L-ascorbic acid trisodium salt (AAP) and generated electron donor-ascorbic acid (AA) with the concomitant increase of the photocurrent, see Figure 4III. The developed photoelectrochemical biosensor reached a linear range from 0.1 to 5000 fM, with a LOD of ~0.033 fM [108].

Other authors have documented the use of enzymatic amplification in the development of immunosensors and genosensors [107,109]. For example, Orozco et al. assembled a genosensor in a sandwich-type format in which a DNA capture probe was immobilized onto the electrode surface by neutravidin-biotin interaction. The capture probe hybridized a DNA target sequence and this one also hybridized a signal probe modified with digoxigenin (Dig) by complementary base pairs. The Dig was able to bind with the antidigoxigenin-HRP complex, which in the presence of a substrate, produced a signal from the enzymatic reaction that was target concentration-dependent. The biosensor was interrogated by chronoamperometry with different RNA target concentrations from toxic algae species, which response was linear in the range from 5 nM to 40 µM, with a LOD of 1.8 nM [107,109].

By using enzymes as signal reporters, Alzate et al. reported a genosensor for the differential diagnosis of zika virus. The genosensor was developed in a sandwich-type format employing anti-Dig-HRP as a signal amplifier, in which changes in target concentration were registered by chronoamperometry. The reported genosensor showed a linear response from 5 to 300 nM with a LOD of 0.7 pM [110], which specificity was tested against genetic material from dengue and chikungunya homologous arboviruses in saliva, serum and urine spiked samples.

Attempts to use two or more enzyme anchoring points have been reported as a strategy to amplify the signal response in the development of biosensors as a way to reach a high sensitivity. In this context, Orozco et al. developed a toxic algae genosensor, in which format they included a signal probe with two Dig anchoring points to link anti-Dig-HRP complex for signal amplification. However, the results showed that the use of two enzyme anchoring points neither increased the signal response to the target nor reduced the background [111].

Similarly, Xue Liu et al. designed an electrochemical biosensor based on the hybridization chain reaction assisted by multiple enzymatic signal response for the detection of a DNA sequence. The biosensor used a thiolated capture probe linked at the surface of a GCE, modified with a tungsten disulfide-MWCNTs composite and electroplated AuNPs (AuNPs/WS_2_-MWCNTs/GCE). The capture probe hybridized the target sequence to form a dsDNA and an auxiliary DNA sequence, at the same time. The auxiliary DNA could hybridize with a biotinylated hairpin 1 (bio-H1) that hybridized with a biotinylated hairpin 2 (bio-H2) triggering a chain reaction. As a result of the hybridization chain reaction, the multiple biotin anchoring points could bind specifically with multiple avidin-HRP conjugates to increase the signal response of the biosensor. After the optimization steps, the performance of the biosensor was tested by DPV, with different target concentrations, showing a linear response from 10 fM to 0.1 nM, with a LOD of 2.5 fM [112].

Biosensors in which enzymes amplified the signal response have utility in the detection of chemical compounds in the food industry. In this context, Wu et al. proposed a nanobioconjugate-based immunosensor for detecting acrylamide, a neurotoxin polymeric molecule used in the chemical industry that can be present in water and food and has a potential carcinogenic effect in humans. The nanobioconjugates were based on gold nanorods (AuNRs) conjugated to a primary antibody (AuNRs-Ab1) and AuNRs conjugated to both a secondary antibody and multi-HRP (HRP-AuNRs-Ab2). The immunosensor was assembled onto a modified GCE first coated with chitosan and later decorated with some SnO_2_-SiC hollow spheres electro-modified with AuNPs. AuNPs served to immobilize an antigen that competed with the analyte in a sample for binding to the AuNRs-Ab1. This nanobioconjugate, in turn, linked the HRP-AuNRs-Ab2 signaling tags to quantify the analyte by CV. The amplification capacity of the nanobioconjugates was evaluated in four different assembled biosensing platforms. The optimal format responded in a linear range from 187 ± 12.3 ng/kg to 104 ± 8.2 μg/kg, with a LOD of 45.9 ± 2.7 ng/kg [113]. The results demonstrated that implementing the AuNRs-Ab1 and multi-HRP-AuNRs-Ab2 nanobioconjugates in the sensing platform improved the resultant electrochemical response significantly and highlighted the importance of multiple enzyme molecules immobilized at nanomaterials for signal amplification in biosensing.

Coming back to biomedical applications, Qu et al. reported an electrochemical biosensor based on hydroxyapatite nanoparticles (HAP) bioconjugated with an antibody and ALP for the detection of BACE1 (the β-site amyloid precursor protein cleaving enzyme 1), which catalyzes the first step in the synthesis of β-amyloid (Aβ) peptides that accumulate in the brain in Alzheimer’s disease. The anti-Aβ and ALP were linked to HAP nanoparticles functionalized with PEI to introduce amino groups, by cross-linking with glutaraldehyde. Otherwise, a solution containing the peptide was dropped onto a GE surface, followed by the addition of a solution containing the target BACE1 (a cleaving enzyme) to test the enzymatic inhibition that the target produced. Finally, the anti-Aβ-HAP-ALP nanobioconjugate was added to the electrode surface, where the anti-Aβ bond specifically with the peptide. The signal response of the biosensor was coming from the hydrolysis of sodium pyrophosphate to produce phosphate ions as a result of the multiple ALP molecules on top of HAP nanoparticles, whose reaction was electrically mediated by molybdate ions (MoO_4_^2−^). The biosensor was tested by SWV, with different target concentrations, showing a linear response from 0.25 to 100 U/mL, with a LOD of 0.1 U/mL [114].

Some researchers have reported the use of electrochemical biosensing platforms by using enzymes as signal amplification reporters to detect microRNA sequences. Zhou et al. described the development of an electrochemical biosensor based on avidin-ALP conjugate as the signaling tag for the specific detection of microRNA-319a. The sensing platform consisted of a thiolated capture probe immobilized onto AuNP-electroplated GEs that hybridized with the miRNA-319a target. A solution containing poly(U) polymerase reaction buffer and biotin-UTP was added to the electrode surface to allow the strand extension reaction mediated by the poly(U)polymerase. Afterward, the avidin-ALP conjugate was added to bind specifically with the multiple biotin molecules as a result of the strand extension. ALP catalyzed p-nitrophenyl phosphate, in a target-concentration dependent manner, which signal response measured by DPV was linear from 10 to 1000 fM, with a LOD of 1.7 fM [115].

Zhang et al. proposed an electrochemical biosensor for the sensitive detection of microRNA based on the construction of a super-sandwich, whose signal response was generated by a DNA-HRP conjugate. The biosensing platform used a thiolated capture probe anchored at a GE. A CHA used two H1 and H2 hairpins and the miRNA-221 target sequence. The target hybridized the hairpin H1 opening the loop and then in the presence of the hairpin H2; the target sequence was displaced to open a new hairpin 1 for n times. The H1-H2 that resulted from the CHA process hybridized the capture probe on top of the GE. Then, a solution containing a DNA sequence (L2) and a DNA-HRP conjugate was dropped onto the electrode surface, hybridizing the dsH1-H2 and leading to a super sandwich with multiple HRP molecules. The signal response of the genosensor was produced by the addition of a TMB solution as an electrochemical probe, containing H_2_O_2_, and measured by amperometry. The developed biosensor response changed with different target concentrations, being linear from 10 to 1000 pM, with a LOD of 0.6 pM [116].

Citing another example, Shuai et al. published an electrochemical biosensor for the specific detection of microRNA-21 based on AuNPs and molybdenum disulfide (MoS_2_) as a transducer platform and ALP enzyme as a signaling tag for the signal amplification. A hollow MoS_2_ microtube-decorated GCE with electroplated AuNPs served as a platform to anchor a thiolated capture probe, which hybridized the miRNA-21 target sequence. By the addition of duplex-specific nuclease (DSN) cleaved the formed dsDNA-miRNA-21 helix. The remaining biotinylated capture sequences bound specifically with the streptavidin-ALP conjugate, producing the signal response. The biosensing platform was tested by DPV, with different target concentrations showing a linear range from 0.1 fM to 0.1 pM, with a LOD of 0.086 fM [117].

Mandli et al. designed an electrochemical sandwich biosensor for the specific detection of microRNA-21 based on the streptavidin-ALP conjugate as a signal amplification reporter. A thiolated capture probe was linked to an AuNP-decorated pencil graphite electrode, which hybridized the miRNA-21 target sequence in a sandwich with a biotinylated signal probe and was further coupled to the streptavidin-ALP conjugate. At optimized conditions, the biosensor was evaluated by DPV, which response was target concentration-dependent in a linear working range from 200 pM to 388 nM, with a LOD of 100 pM [118].

### 4.3. Nanomaterials for Enhanced Signal Amplification

Metallic nanomaterials such as noble metals are widely employed as transducer platforms in the development of biosensors. They not only have enhanced physical, chemical and electrochemical properties but are resistant to corrosion and oxidation processes and environments. Noble metals, including Au, Pt, Pd, and Ag, among others, have found application as reporters for signal amplification in biosensing. Their outstanding features depend on their size and shape and are related to their high surface area and stability. Noble metals such as Ag and Au exhibit strong surface plasmon resonance as a result of a collective coherent oscillation that generates electron bands, of high capacity for electron transfer and exceptional utility in biosensor development.

Bo Bing et al. used AuNP-DNA nanobioconjugates to develop a signal amplification platform for detecting microRNA. The amplification was based on the construction of a bridge between AuNPs-DNA nanobioconjugates and DNA strands. The nanobioconjugate bridge hybridized a fragment of a capture probe, which resulted in the cleavage of the duplex capture-target DNA by a nuclease enzyme. A Ru complex produced the electrochemical response in the biosensing platform as an electroactive probe adsorbed in the DNA of the nanobioconjugate bridge. The biosensor performance was interrogated by chronocoulometry, showing a linear range from 0.01 fM to 10 pM, with a LOD of microRNA down to 6.8 aM [86].

Many other authors have built electrochemical biosensing platforms for the specific detection of DNA or microRNA sequences. Shi et al. illustrated the development of a photoelectrochemical DNA biosensor based on an inorganic-organic composite for dual signal amplification. A carboxylated DNA probe was immobilized on top of meso-tetra (4-carboxyphenyl) porphine (TCPP)-decorated CdTe QDs by the EDC/NHS coupling reaction between the amino groups from QDs and carboxyl groups from DNA. An ITO-based electrode served as a transducer platform. It was decorated with ZnO NPs, which were further coated with CdS nanocrystals by Cd^2+^-S^-^ interaction forming a hetero ZnO/CdS nanostructure. A 3′-thiolated and 5′phosphorylated DNA hairpin was then immobilized onto the electrode surface to hybridize the target DNA, followed by the addition of exonuclease (λ-Exo) that selectively cleaved ds-DNA with one 5′-phosphorylated end in the 5′ to 3′ direction. The resultant DNA fragment on top of the electrode hybridized the DNA sequence from the nanobioconjugate, see Figure 5I. The biosensor was tested by the photocurrent changes in a target concentration-dependent manner, which response was linear from 0.1 fM to 5 pM, with a LOD of 25.6 aM [119].

Chen et al. developed an electrochemical biosensor for the specific detection of microRNA-21 based on DNA-functionalized AuNPs as labels for signal amplification. After synthesizing the AuNPs by a reduction method, they were bioconjugated with both thiolated DNA and hairpin (H1) probes (H1-AuNP-DNA). The transducer platform consisted of a GCE decorated with carbon sphere-molybdenum disulfide (CS-MoS_2_) and coated with electroplated AuNPs. A thiolated DNA capture probe was then immobilized onto the electrode surface to hybridize the DNA probe from the nanobioconjugates. Next, the target microRNA-21 was added to the electrode surface to allow the hairpin H1 to open, followed by the addition of a second biotinylated hairpin H2, which displaced the target microRNA-21. The above-catalyzed hairpin cycle was repeated several times, as shown in Figure 5II. Finally, the streptavidin-HRP bond to the biotin from the H2 and the signal response was recorded by DPV. The biosensing platform was tested with different microRNA-21 target concentrations being linear in a range from 0.1 fM to 0.1 nM, with a LOD of 16 aM [120].

Liang et al. developed an ultrasensitive biosensor based on a catalytic nanoprobe of copper-metal frameworks (CuNMOF) decorated with PtNPs and biofunctionalized with HRP for the specific detection of microRNA-151. The CuNMOF was prepared by a hydrothermal method with copper nitrate and 2-aminoterephthalic acid as precursors, followed by coating with PtNPs by reducing Pt ions with NaBH_4_. Both, a signal probe (ssDNA) and HRP were coupled to the framework to obtain the resultant CuNMOF@PtNPs/HRP/ssDNA nanobioconjugates. The transducer was built by linking a thiolated tetrahedral DNA capture probe containing a hairpin to a GE. The electrode surface hybridized the target sequences by a toehold strand displacement reaction (TSDR) with the hairpin on top of the tetrahedron. At this point, the as-prepared nanobioconjugate was dropped onto the electrode for signal amplification, which interaction was followed by SWV, being target concentration-dependent and linear in a range from 0.5 to 100 pM, with a LOD of 0.13 fM [121].

As depicted in Figure 5III, Han et al. assembled an ultrasensitive electrochemical biosensor based on ssDNA-AuNPs and ssDNA-CNT as signaling tags for the detection of DNA. Two different thiolated DNA probes were chemisorbed on top of the AuNPs surface. Whereas one of the sequences was specific for the target, the other one hybridized with DNA from the DNA-CNT nanobioconjugate, prepared by covalent coupling aminated DNA to carboxyl moiety-coated CNTs by the EDC/NHS chemistry. Aminated capture probe sequences were anchored at dopamine grafted GE to hybridize the target sequence. Afterward, the sequential addition of the DNA-AuNP and DNA-CNT nanobioconjugates complete the biosensing assay. Finally, the target DNA was tested by linear sweep voltammetry (LSV), where the signals were target concentration-dependent in a linear range from 0.1 pM to 10 nM, with a LOD of 5.2 fM [122].

Signaling tags based on nanobioconjugates for electrochemical biosensing platforms have been produced by using biomolecules different from DNA/RNA sequences. For this purpose, Xu et al. reported an electrochemical peptide-based biosensor for the detection of matrix metalloproteinase 2 (MMP-2), with Pt/Pd-CeO_2_ nanospheres bioconjugated with streptavidin and thionine (Th), as signaling tags, see Figure 5IV. CeO_2_ nanospheres were synthesized by a hydrothermal method and decorated with Pd and PtNPs through the carboxyl groups rich lysine linker. Afterward, a solution containing Th was added to the hybrid material, followed by the addition of streptavidin to obtain the streptavidin/Th/Pt/Pd/CeO_2_ nanobioconjugate. The transducer was a thiolated peptide with a terminal cysteine residue (P1) anchored at an electroplated AuNP-GCE surface. Then, the electrode surface was treated with MMP-2, a class of zinc-dependent endopeptidase that specifically recognized and cleavaged the P1. Finally, the nanobioconjugate bond specifically the not cleaved biotinylated peptide by the biotin-streptavidin interaction. At optimal conditions, the biosensor tested by DPV was MMP-2 target-dependent and responded linearly in concentrations ranging from 0.1 pg/mL to 10 ng/mL, with a LOD of 0.078 pg/mL [123].

As illustrated in Figure 5V Zhang H et al. designed a biosensing platform based on AuNPs functionalized with streptavidin (AuNPs-strep) for the specific detection of microRNA-21. The biosensor platform was constructed on GEs for the immobilization of thiolated DNA hairpin. The hairpin opened in the presence of the DNA target sequence by hybridization with the complementary bases. The hybridized sequence was cleavaged by a nuclease enzyme, which was specific for dsDNA. The cleavage process left a single portion of the hairpin to hybridize with a biotinylated DNA signal sequence. The biotin from the signal probe bound specifically with streptavidin from the strep-AuNP nanobioconjugates. The streptavidin molecules on top of AuNPs allowed the binding of many biotin-HRP complexes that, in the presence of a mediator, produced a redox reaction directly correlated with the microRNA concentration. The biosensor performance was interrogated by amperometry, which response was linear from 0.1 fM to 100 pM, with an ultrasensitive LOD down to 43.3 aM [124].

Otherwise, Shiwei Zhou et al. reported an electrochemical immunosensor based on a nanobioconjugate synthesized with solid silica (SiO_2_) NPs decorated with QDs and bioconjugated with a secondary antibody for the simultaneous detection of B-cell lymphoma 2 (Bcl-2) and Bcl-2-associated X protein (Bax). SiO_2_ NPs were treated separately with a solution containing poly(diallyldimethylammonium chloride) (PDDA) and PEI. The PDDA-modified SiO_2_ and PEI-modified SiO_2_ NPs were dispersed into solutions containing CdSeTe@CdS QDs and silver nanoclusters (AgNCs), respectively, and treated with solutions containing the anti-Bcl-2 and anti-Bax antibodies, respectively both anchored by the EDC/NHS coupling reaction. Primary anti-Bcl-2 and anti-Bax antibodies were covalently coupled to an rGO-decorated GCE, which interacted with solutions containing the Bcl-2 and Bax antigens and the anti-Bcl-2- and anti-Bax-containing nanobioconjugates. The antigens were determined by ASV, by measuring the oxidation peak currents of Cd and Ag. At optimal conditions, the immunosensor was antigen-concentration dependent in a linear range from 1 to 250 ng/mL, with a LOD ~0.5 fM for Bcl-2 and Bax, respectively [125].

You et al. assembled an ultrasensitive biosensor for the detection of a glycoprotein based on SiO_2_ NPs decorated with AuNPs and bioconjugated with ferrocenylhexanethiol (Fc) and mercaptophenylboronic acid (MPBA) as signaling tags. SiO_2_ NPs were modified with APTES, coated with AuNPs and further functionalized with the Fc and MPBA thiolated molecules (Fc/MPBA/AuNPs-SiO_2_). The transducer was prepared by decorating a chitosan-modified GCE with AuNPs-rGO. Then, MPBA was linked to the electrode surface to bind the target glycoprotein, which in turn, linked the Fc/MPBA/AuNPs-SiO_2_ nanobioconjugate in a sandwich-type glycobiosensor. The concentration of glycoprotein was estimated by DPV. At optimized conditions, the biosensor response was linear from 1 pg/mL to 250 ng/mL, with a LOD of 0.57 pg/mL [126].

Li et al. described an electrochemical biosensor based on streptavidin-Cd(II) QD bioconjugates for the detection of telomerase activity. A thiolated DNA capture probe was anchored at the surface of a GE, followed by repeated telomerase extension by the addition of specific telomerase primers and biotinylated nucleotides, thus getting multiple biotin anchoring points. A solution containing the streptavidin-QD nanobioconjugates was added to the electrode surface to bind specifically with the multiple biotin molecules onto the elongated DNA sequence. Finally, nitric acid was added to dissolve the Cd(II) from the nanobioconjugates and the signal response interrogated by SWASV. The developed biosensing platform was tested with different concentrations of telomerase extracts reaching a linear working range from 1 to 105 cells, with a LOD of 0.37 cells [127].

DNA biomarkers on top of nanomaterials not only allow the detection of complementary DNA sequence but the detection of glycoproteins. As a model, Li et al. proposed an ultrasensitive electrochemiluminescence biosensor based on nanobioconjugates for DNA target recycling and a DNA dendrimer to act as a signaling tag for the specific detection of laminin (LN), a glycoprotein involved in many important biological functions and used as a biomarker of liver fibrosis. The dendrimer was developed by self-assembling two DNA complementary sequences and using a trimeric cross-linker (TMEA) to generate a Y-shape DNA arrangement. The DNA dendrimer was then functionalized with *N*-(aminobutyl)-*N*-(ethylisoluminol) (ABEI) and conjugated with doxorubicin (Dox), as intercalator of dsDNA (ABEI-Dox). Otherwise, AuNPs were conjugated with two different DNA sequences (DNA1 and DNA3) that served for the protein conversion. While the carboxylated DNA1 was functionalized with a primary antibody (Ab1) by the EDC/NHS coupling reaction, the DNA3 hybridized a complementary DNA sequence (S2). A solution containing DNA2 functionalized with a secondary antibody (Ab2) was dropped onto the nanobioconjugate (AuNPs/DNA1-Ab1/DNA3-S2) to displace the S2 DNA sequence. Then, a solution containing the LN that bond specifically to the antibodies was added to mix with the AuNPs/DNA1-Ab1/DNA3-S2 and DNA2-Ab2 solutions. Subsequently, exonuclease III (EXO II) was added to the nanobioconjugate to digest the DNA3/DNA2, thus releasing DNA for further hybridization, many times. The transducer platform used a capture probe S1 immobilized onto an AuNP-electroplated GCE, functionalized with L-cysteine. The capture probe S1 hybridized the S2 DNA sequences that resulted from the hybridization cycling process. Finally, S2 sequences hybridized the DNA sequences from the dendrimer to complete the biosensing platform. The developed biosensor was interrogated by photocurrent changes from different target concentrations, which response was linear from 0.1 pg/mL to 100 ng/mL, with a LOD of 0.0661 pg/mL [128].

A series of electrochemical biosensing platforms are shown as follow based on nanomaterials of different chemical composition to detect specific targets at ultralow concentrations. Li et al. published an electrochemical biosensor based on single-wall carbon nanohorns doped with platinum nanoclusters (SWCNHs/PtNC) as the signaling tag for the detection of α2,6-sialylated glycans, a potential biomarker for tumors, see Figure 5VI. The hybrid material was prepared by a hydrothermal method and further coupled with 3-aminophenylboronic acid (APBA) by the EDC/NHS chemistry. A GCE coated with streptavidin and gold nanorods (streptavidin-AuNR) was used as a platform to immobilize a biotinylated antibody that specifically recognized the α-2,6-sialylated glycan target molecules. Finally, the APBA-SWCNH/PtNC nanobioconjugates were linked to the platform through the glycan-boronic acid interaction. The response of the developed biosensor was interrogated by two methods, i.e., a label-free platform (without nanobioconjugates) by amperometry and a labeled platform (by using nanobioconjugates) by DPV. Both platforms were target analyte concentration-dependant. The response was linear in a range from 1 fg/mL to 100 ng/mL, with a LOD of 0.69 ng/mL and from 5 ng/mL to 5 μg/mL, with a LOD of 0.50 ng/mL for the labeled and label-free platforms, respectively. The results demonstrated the higher sensitivity of biosensors developed with nanobioconjugates as compared with the same biosensor without them [129].

Wang et al. designed an electrochemical aptasensor based GO decorated with AuNPs and bioconjugated with an aptamer (Apt2) for the specific detection of Pb^2+^. A thiolated aptamer Apt2 was immobilized onto a hybrid material based on GO decorated with AuNPs, which were previously synthesized by reduction with NaBH_4_ (GO@AuNPs-Apt2). The sensing platform consisted of a thiolated aptamer Apt1 anchored at the surface of a GCE decorated with the GO@AuNPs. After adding the GO@AuNPs-Apt2 nanobioconjugate, Apt1 hybridized Apt2, which was then cleavaged by Pb^2+^. The biosensor response was evaluated by amperometry, being linear in a range from 5 pM to 1 µM, with a LOD of 1.67 pM [130].

Han et al. reported a MoS_2_-AuNP-based electrochemical aptasensor for the specific detection of zearalenone (ZEN) and fumonisin B1 (FB1) mycotoxins, which signal was amplified by an AuNP-ssDNA bioconjugate. Two different sequences of DNA (CP1 and CP2) were immobilized on top of AuNPs and then treated with Thi and 6-(Ferrocenyl) hexanethiol (FC6S), respectively. On the other hand, two different thiolated aptamers (AP1 and AP2) were anchored at a MoS_2_-AuNP-decorated GCE to interact with the ZEN and FB1 target molecules, respectively. The ferrocenyl-containing nanobioconjugates linked then to the target molecules, thereby producing signals by DPV that were correlated with changes in their concentrations. The dual electrochemical biosensor responded in a linear range from 1 µg/mL to 10 ng/mL and 1 µg/mL to 100 ng/mL for ZEN and FB1, respectively, with a LOD of 0.5 µg/mL in both cases [131].

Liu et al. illustrated the development of a biosensor based on AuNPs bioconjugated with polyadenine oligonucleotides (polyA-ODNs) to construct a 3D-controlled rolling DNA for the detection and signal amplification of adenosine molecule as a model analyte. AuNPs, synthesized by a reduction method, were functionalized with polyA-ODNs that contain a cleavage point and ferrocene in one of its extremes (AuNPs-polyA-ODNs). A tetrahedron DNA capture probe, having a free ssDNA on top of its structure, was immobilized at a GE for the specific interaction with the AuNPs-polyA-ODN nanobioconjugate. Adenosine was then added to the electrode surface to interact with the polyA block and ferrocene to form a cleavage point activated by the presence of the Mg^2+^ cofactor in a process repeated n-times. After the optimization of the involved parameters, the biosensor response, interrogated by DPV, was linear from 0.5 nM to 1.5 μM and exhibited a LOD of 0.17 nM [132].

### 4.4. Electroactive Complex and Dyes as Amplifiers of the Signal Response

The use of complexes and dyes as signal amplifiers is a strategy widely employed in the biosensing field [86,133,134], where the electroactive complex or dye commonly intercalate (or adsorb) in the dsDNA through π-π stacking (or ionic interactions) [135]. For example, whereas dyes such as MB binds specifically with guanine bases from DNA due to its planar structure [136], the electroactive ruthenium complex adsorbs on the DNA sequences through DNA phosphate backbone-Ru^3+^ electrostatic interactions [75,137]. The accumulation of numerous molecules of MB and Ru^3+^ in the vicinity of the sensing surface and in between the DNA sequences produce a dramatically amplified response with no need for other mediators [75].

In this context, Cheng et al. described a biosensing platform to detect microRNA with nanobioconjugates based on streptavidin-Cd^2+^-modified titanium phosphate (TiP) NPs as signal tags, Figure 6I. A transducer gold platform was modified with an AuNP-rGO hybrid material to increase the surface area. A capture probe 1 chemisorbed on top of AuNPs from the transducer surface hybridized with a portion of the target DNA sequences A biotinylated capture probe 2 that hybridized with another portion of the target also bond with the streptavidin-Cd^2+^-TiP nanobioconjugate. The Ru(NH3)_6_^3+^ complex, adsorbed on the DNA bases, was the electron transfer mediator responsible for the resultant amplified electrochemical signal. The signal response of the as-developed biosensor was interrogated by SWV, being concentration-dependent in a linear range from 1 aM to 10 pM, with a LOD of 0.76 aM [133].

Li et al. proposed a biosensor for DNA detection employing DNA-AuNPs/Ir nanobioconjugates as signaling tags and the [(ppy)_2_Ir(dcbpy)]^+^ PF^6−^ complex as an electrochemical reporter. A capture probe immobilized onto an Au electrode surface hybridized with the target, which in turn hybridized with the nanobioconjugate. The complex was coupled onto the AuNP-DNA nanobioconjugate surface by using cysteamine as a bio-barcode where the amine group served to coordinate with the Ir complex. The signal response of the biosensor was interrogated by current intensity changes upon the target concentration variation. The authors compare the biosensor performance of iridium and ruthenium complexes. The results showed that the use of iridium complex as the label was better for signal amplification, in a linear range from 2 fM to 1 pM, with a LOD of 0.86 fM, lower than that from the ruthenium one and with 400-fold more sensitivity [138].

By using another electroactive complex, Hui Zhang et al. designed a sensitive electrochemical biosensor based on a DNA-AuNP bioconjugate for the specific detection of DNA (cytosine-5)-methyltransferase 1 (DNMT1), which activity is crucial in cancer research, see Figure 6II. The AuNPs were synthesized by a reduction method and further functionalized with two different thiolated DNA probes named S2 and S3, being S2 complementary to the target sequence. The capture probe S1 having a preferred DNMT1 methylation site was linked to a GE and let to interact with the AuNPs/S2/S3 through the S1-S2 hybridization. Then, a solution containing the gene-specific methylation restriction enzyme BssHII was added to the electrode surface to cleave the ds-S1-S2 probes. Finally, the signal response of the biosensor was amplified by the Ru^3+^ complex that was adsorbed at the phosphate backbone from DNA probes through electrostatic interactions. Finally, the biosensor was tested by DPV with different target concentrations, which signal response was linear in a range from 1 to 40 U/mL, with a LOD of 0.3 U/mL [139].

Similarly, Shao et al. published an electrochemical biosensor for the detection of microRNA-21 based on AuNPs functionalized with MoS_2_ sheets as signaling tags assisted by a ruthenium complex as signal response amplifier as shown in Figure 6III. The AuNP@MoS_2_ was synthesized and decorated with a thiolated signal probe DNA2, and a thiolated capture probe DNA1 was immobilized at the surface of AuNP@MoS_2_-coated GCE. A solution containing the miRNA-21 target was added to the electrode surface to hybridize in between the capture and signal probes. Finally, a solution containing the Ru^3+^ complex was dropped onto the electrode surface to interact specifically with the phosphate groups from the DNA backbone to produce the signal response. The current response was evaluated by EIS and DPV, which results demonstrated that the biosensor was target concentration-dependent in a linear range from 10 fM to 1 nM, with a LOD of 0.78 and 0.45 fM for DPV and EIS, respectively [140].

Cui et al. reported an ultrasensitive electrochemical biosensor based on DNA-functionalized AuNPs for the polynucleotide kinase (PNK) detection, an enzyme involved in nucleic acid metabolism and cellular response in DNA damage, see Figure 6IV. A thiolated DNA 1 probe linked to a GE hybridized with a DNA 2-AuNP nanobioconjugate. The electrode was treated with a PNK solution to induce the phosphorylation of the DNA strands, followed by the addition of ATP to inhibit the PNK activity. Finally, the enzyme lambda exonuclease that cleaved the DNA probes in the 5′-phosphorylated DNA end was added, followed by a ruthenium complex as a signaling reporter. In the absence of PNK, the DNA probe keep unaltered at the electrode surface, thus retaining several ruthenium molecules for the electrochemical signal response. The biosensor was evaluated by DPV, which response was PNK concentration-dependent in a linear range from 0.001 to 10 U/mL, with a LOD of 7.762 × 10^−4^ U/mL [141].

Some authors have employed dyes as an alternative to the use of electroactive ruthenium complexes to amplify the signal response of electrochemical biosensors. For this purpose, Miao et al. assembled an ultrasensitive electrochemical biosensor based on AuNPs functionalized with ssDNA and assisted by MB as signaling reporter, for the specific detection of the tyrosine kinase-7 protein, a cancer biomarker-associated cell membrane protein, see Figure 6V. AuNPs were synthesized by a reduction method followed by immobilization of thiolated DNA probes. The biosensing platform was a thiolated aptamer linked to a GCE previously functionalized with Nafion and decorated with AuNPs through electrostatic interactions. In the absence of the target, a helper probe hybridized in between the aptamer and DNA from the nanobioconjugates. MB was then intercalated in between the dsDNA producing a signal response. In the absence of the target, the aptamer changed conformation as a result of the aptamer-target protein interaction. Such conformation change prevented the nanobioconjugate to be linked to the platform and the concomitant MB intercalation. Then, the signal response tested by DPV was dependent on target concentration, which was inversely proportional but linear from 1 pm to 100 pM and from 100 pM to 1 nM, with a LOD of 372 fM [142].

Liu et al. illustrated the development of a nanobioconjugate-based enzyme-free electrochemical biosensor consisting of an AuNP-coated Fe_3_O_4_/CeO_2_ hybrid material decorated with DNA sequences for the detection of microRNA-21, see Figure 6VI. Fe_3_O_4_/CeO_2_ NPs were obtained by a hydrothermal method, further coated with AuNPs and functionalized with a thiolated DNA sequence (S1) to obtain the Fe_3_O_4_/CeO_2_@Au-S1 nanobioconjugate. A hairpin H1 was attached to an AuNP-coated GCE to link the target sequences that were then displaced by a H2, which process was repeated n times. The Fe_3_O_4_/CeO_2_@Au-S1 nanobioconjugate was then added to hybridize with a portion of the H1 to complete the biosensing assay. Finally, MB dye intercalated in between the dsDNA, thus producing a signal by DPV that was correlated to the target concentration. The biosensor response was linear, ranging from 1 fM to 1 nM, with a LOD of 0.33 fM [143].

Lin et al. developed an immunosensor for the detection of GP73, a protein that plays a key role in the sorting and modification of proteins from the endoplasmic reticulum. The immunosensor was based on a biotinylated DNA tetrahedron as an immobilization probe for anchoring a primary antibody through the biotin-streptavidin interaction. The antibody was specific for the antigen Golgi protein 73, which at the same time bound with a secondary antibody covalently modified with Ru^3+^ complex. MB intercalated in between the dsDNA from the tetrahedron, thus giving an internal reference signal. The Ru^3+^ covalently coupled to the secondary antibody was responsible for the current signal response. The immunosensor response was linear, ranging from 15 pg/mL to 0.7 ng/mL, with a LOD of 15 pg/mL. The device was able to detect the antigen present in real samples [144].

The above section illustrated the development of different nanobioconjugates as signaling tags for the enhanced signal amplification in a variety of biosensing platforms. These nanobioconjugates based on nanomaterials have been assembled to increase the signal response of biorecognition events. Most of the reported electrochemical biosensors that use nanobioconjugates to increase the signal response are a proof-of-concept for testing target biomolecules associated with pathogens and other biomedical and environmental applications that include the detection of heavy metals among other uses. However, some other reports go a step further towards testing real samples, thus demonstrating the capabilities of signal amplification response of the proposed biosensors and their enormous potential for sensing in real scenarios. Overall, nanobioconjugates, as components of electrochemical biosensing platforms, have demonstrated their potential to be used in the development of a routine assay for a variety of target analytes. Table 3 summarizes selected illustrative research works in which nanobioconjugates were assembled in electrochemical biosensing platforms for the improved signal amplification of biorecognition events.

## 5. Current Challenges and Future Perspectives in Nanobioconjugates Development

We have reviewed several biosensing reports that include nanobioconjugates for signal amplification in multiple formats. They have outstanding performance and the potential to achieve high sensitivity and ultralow limits of detection in nanobioconjugate-based assays. Their features are compatible with multiplexing and miniaturization, as well as portability and low volume of samples and reagents. However, there are still some challenges to face before implementing such reporters in biosensors in a real scenario. For example, biomarkers are commonly present in biological fluids in extremely low concentrations [145], so their detection and monitoring require highly sensitive devices.

The use of nanostructured materials in the development of electrochemical biosensing platforms offers the opportunity of a wide range of modifications with different bioreceptors and thus specific detection of a myriad of target biomolecules. All parameters involved in the nanobioconjugates development need to be systematically optimized and standardized before being incorporated in electrochemical biosensing platforms for the detection of the target molecules with high sensitivity and specificity. In this context, nanobioconjugates stability must be well established to ensure nanobioconjugates performance and reproducibility of biosensors. Parameters influencing the long term stability of nanobioconjugates such as nanomaterial geometric shape [146], ionic strength [147], pH [148], and temperature [149] need to be optimized. The need to strictly control the number of affinity biomolecules in a nanobioconjugate to avoid unwished phenomena is still a challenge. An excessive number of biomolecules linked onto nanostructured surfaces may hinder their biochemical activity, which alters their targeting ability by cross-linking with other molecules [17,18]. Conversely, multivalent interactions and associated cooperative binding become insignificant whether nanostructures have a lower number of biomolecules attached to them [17].

Yet, nanobioconjugates have tremendous potential for the development of biosensing platforms and devices of superior performance, with an excellent capacity for enhancing the signal amplification processes of biorecognition events. They have many advantages as compared with conventional assays. Although conventional assays are ordinarily used, many of them are time-consuming and require robust instrumentation, which hinders timely and accurate target detection and quantification [150]. Advantages of nanobioconjugate-based assays include their relatively lower cost and faster analysis time, and the fact that they do not require expensive equipment and there is no need for well-trained personal. The aforementioned features make such nanoplatforms hold the potential to be implemented for analysis in place, even in remote settings.

Two-dimensional nanomaterials are emerging nanomaterials of enhanced physical, chemical and optical properties as compared to their bulk counterparts [151], which make them promising for the development of nanobioconjugates. For instance, their high surface area allows for hosting thousands of biomolecules in a proper nano-environment that promotes the stability and activity of biomolecules [151,152]. Implementation of 2D nanomaterials in nanobioconjugate assemblies opens opportunities towards keep increasing sensitivity and stability and decreasing the LOD of bioassays where they are assembled.

Overall, nanobioconjugates offer the possibility to detect different target biomolecules with high specificity and sensitivity in less time and in a straighter forward way as compared with standard assay methods. Such unique features, along with versatility, accurate quantification, and amenability for multiplexing and miniaturization, are paving the way towards the development of new enhanced nanobioconjugate-based device alternatives. Progress remains to be made for positioning this technology in the market. It requires joined efforts from cross-disciplinary fields that involve nanochemistry and nanobiotechnology. However, it is clear that nanobioconjugates are at the forefront of research in many fields, not only as reporters in signal amplification in biosensors, but as nanocarrires for targeted drug delivery and contrast agents in biomedical imaging among many others, always searching for novel and new opportunities depending on the final application.

## 6. Concluding Remarks

This review illustrated a variety of strategies for assembling and characterizing nanobioconjugates based on the conjugation of nanomaterials with biomolecules. It focused on describing the high potential of the nanobioconjugates for their application in biosensing systems in which they can dramatically amplify the signal response of biorecognition events. The review described the conditions to have nanobioconjugates with high colloidal stability. It showed bioconjugation strategies to promote the conjugation of an appropriate number of well-oriented and active biomolecules with nanomaterials that prevent biomolecules lose their activity. The resultant nanobioconjugates were implemented in different biosensor formats. For example, DNA-based nanobioconjugates were applied in genosensors development to detect DNA, RNA, or microRNA associated with cancer and infectious diseases, among others [86,104,153,154,155]. Antibody-based nanobioconjugates were used to develop immunosensors to detect the level of antigens [156], and enzyme-based nanobioconjugates to develop enzymatic sensors to monitor glucose levels, DNA, and antigens, among others [157]. Glycan-based nanobioconjugates to design biosensors for glycoproteins detection [126]. The resultant platforms had enhanced features and improved performance regarding the concomitant simple format counterparts. Overall, the summarized examples showed plenty of biosensing formats based on nanobioconjugates for signal amplification, but also revealed their tremendous potential for many other applications that involve cross-disciplinary fields at the nanoscale level.

## Figures and Tables

**Figure 1 molecules-25-03542-f001:**
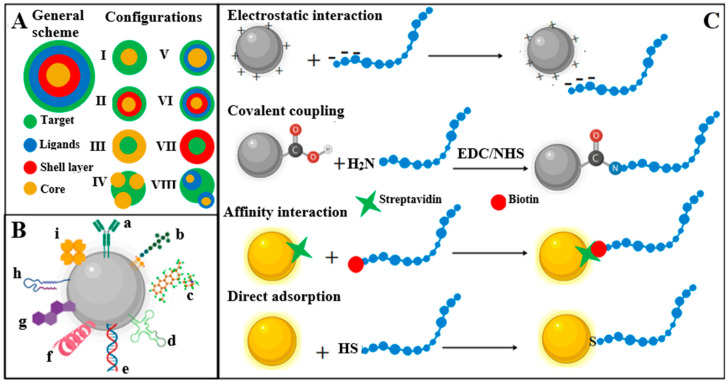
Schematic illustration of a nanobioconjugate. (**A**) Eight different nanobioconjugate configurations (I) Direct coupling of biomolecules onto a NP surface, (II) biomolecule immobilization on top of a shell layer of an hybrid NP, (III) biomolecule encapsulation, (IV) biomolecules decorated with smaller size NPs, (V) biomolecule immobilization by secondary interactions via ligands onto a NPs surface, (VI) biomolecule immobilization via secondary interactions with a linker on top of a shell layer of an hybrid NP, (VII) biomolecules entrapped into hollow NPs, (VIII) biomolecules decorated with smaller size NPs via secondary interactions, (**B**) Nanoparticle interaction with different biomolecules, including (a) antibodies, (b) glycans, (c) molecules, (d) aptamers, (e) dsDNA, (f) ssDNA, (g) steroids, (h) peptides and (i) proteins, (**C**) Different types of interactions between nanomaterials and biomolecules.

**Figure 2 molecules-25-03542-f002:**
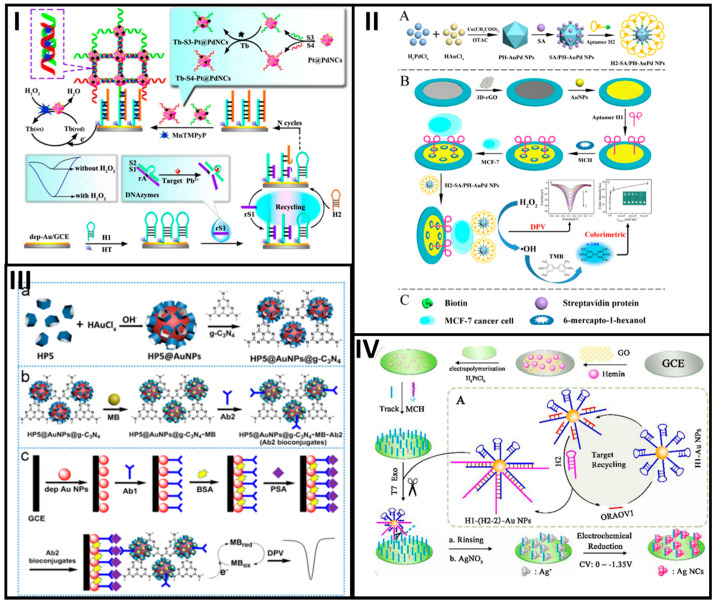
Illustrative examples of electrochemical biosensors developed with noble nanomaterial-based nanobioconjugates. (**I**) Biosensor developed for the sensitive detection of Pb^2+^ based on the use of DNA functionalized (Pt@PdNCs) hybrid material, reprinted from reference 26, copyright Elsevier 2016. (**II**) Polyhedral AuPd alloy nanoparticles (PH-AuPd NPs) conjugated with a DNA aptamer for the specific detection of MCF-7 cells, (A) Synthesis of PH-AuPd NPs and H2/SA-PH-AuPd nanobioconjugation, (B) assembly of the electrochemical cytosensor, (C) some components of the electrochemical platform, reprinted from reference 29, copyright Elsevier 2018. (**III**) Electrochemical biosensor developed with hydroxyl pillar[5]arene@AuNPs@g-C_3_N_4_ hybrid nanomaterial biofunctionalized with anti-PSA aptamer for the detection of PSA, (a) synthesis of the HP5@AuNPs@g-C_3_N_4_ hybrid nanomaterial, (b) HP5@AuNPs@g-C_3_N_4_ functionalization with MB and Ab2, (c) assembly of the electrochemical immunosensing platform, reprinted from reference 30, copyright Elsevier 2018. (**IV**) Electrochemical biosensor developed for the specific detection of ORAOV1 based on DNA nanobioconjugates, (A) catalytic hairpin recycling for target ORAOV1 molecular amplification, reprinted from reference 31, copyright Elsevier 2020.

**Figure 3 molecules-25-03542-f003:**
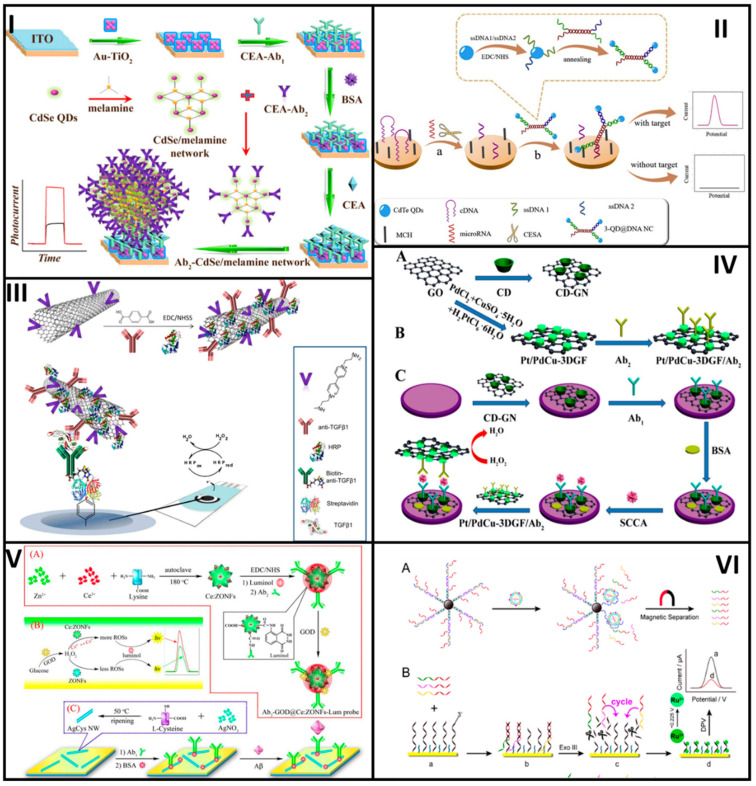
Examples of electrochemical biosensors developed with nanomaterial-based nanobioconjugates. (**I**) Biosensor developed with CdSe-QD-melamine networks for the detection of the CEA, reprinted from reference 41, copyright Elsevier 2016. (**II**) Biosensor based on a concatemer-QDs linked to a DNA-aptamer for signal amplification in the ultrasensitive detection of cancer cells, (a) hybridization process between capture DNA and microRNA probes followed by dsDNA cleavage by cyclic enzymatic signal amplification, (b) hybridization process between capture DNA fragments and 3-QD@DNA nanobioconjugates, reprinted from reference 42, copyright Elsevier 2019. (**III**) Electrochemical immunosensor for the determination of transforming growth factor β1 (TGF-β1) based on CNT bioconjugates, reprinted from reference 54, copyright Elsevier 2017. (**IV**) Electrochemical immunosensor for the detection of squamous cell carcinoma antigen (SCCA) based on two different nanobioconjugates using graphene, reprinted from reference 55, (A) functionalization of GO with β-cyclodextrin (B) development of Ab2-Pt/Pd/Cu-GO nanobioconjugates, (C) assembly of the nanobioconjugate-based electrochemical immunosensor, copyright Elsevier 2015. (**V**) Immunosensing platform for the detection of amyloid-β protein based on nanobioconjugates, (A) Synthesis of Ab2-COD@Ce:ZONF-luminol nanobioconjugates, (B) ECL mechanism proposed for the signal generation and (C) Synthesis of AgCys nanowires for the electrode modification and development of the electrochemical sensing platform, reprinted from reference 61, copyright American Chemical Society 2016. (**VI**) Electrochemical biosensor based on an aptamer-DNA functionalized-MNPs for the tumor exosomes detection, (A) aptamers-exosomes conjugation and release of multimessenger DNAs, (B) detection of released multimessenger DNA by the electrochemical platform, reprinted from reference 68, copyright American Chemical Society 2018.

**Figure 4 molecules-25-03542-f004:**
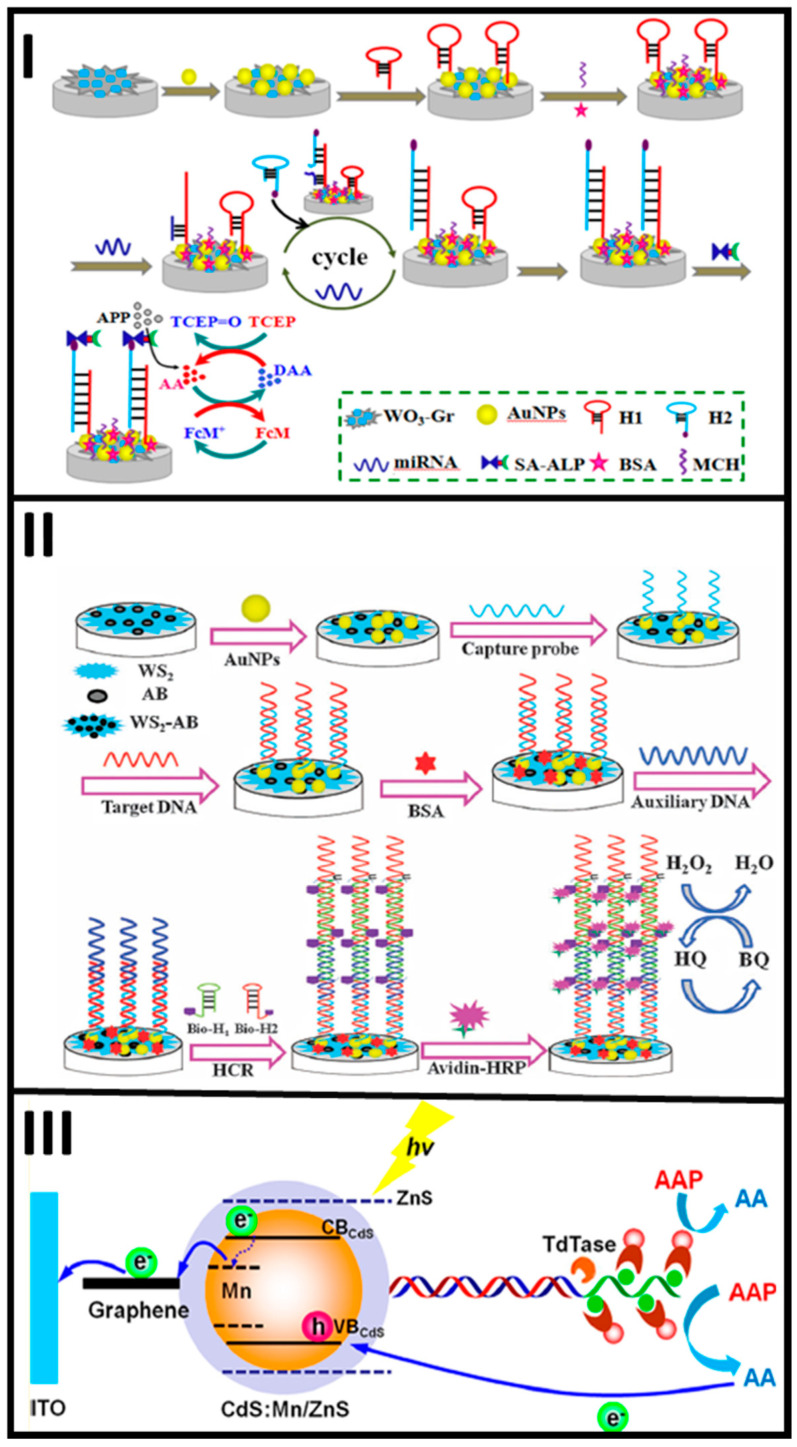
Schematic representation of biosensors based on molecular strategies. (**I**) Electrochemical microRNA biosensor based on the catalyzed hairpin assembly as signal amplification, reprinted from reference 100, copyright Elsevier 2016. (**II**) Schematic design of the biosensor and amplification via a hybridization chain reaction, reprinted from reference 105, copyright Royal Society of Chemistry 2016. (**III**) Electrochemical biosensor developed by using QD nanobioconjugates, reprinted from reference 108, copyright American Chemical Society 2015.

**Figure 5 molecules-25-03542-f005:**
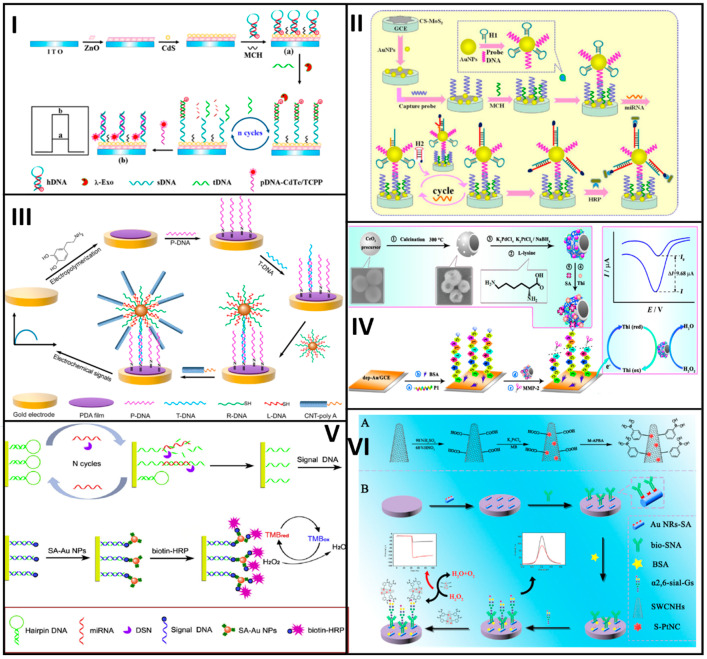
Schematic illustration of an electrochemical biosensor based on nanomaterial-based nanobioconjugates for signal amplification. (**I**) Electrochemical biosensing platform for the specific detection of DNA or microRNA based on QDs, (a) immobilization of thiolated hairpin DNA onto electrode surface, (b) hybridization of remaining fragment of hairpin DNA and pDNA-CdTe/TCPP conjugates, reprinted from reference 119, copyright American Chemical Society 2016. (**II**) Electrochemical biosensor for the specific detection of microRNA-21 based on DNA-functionalized AuNP nanobioconjugates, reprinted from reference 120, copyright Elsevier 2018. (**III**) Ultrasensitive electrochemical biosensor based on two nanobioconjugates, ssDNA-AuNP and ssDNA-CNT as signaling tags for the detection of DNA, reprinted from reference 122, copyright American Chemical Society 2020. (**IV**) Electrochemical biosensor based on Pt/Pd-CeO_2_ nanospheres bioconjugated with streptavidin and Th as signaling tags, for the detection of matrix metalloproteinase 2 (MMP-2), reprinted from reference 123, copyright Elsevier 2016. (**V**) Nanobioconjugates based on AuNPs functionalized with streptavidin for the specific detection of microRNA-21, reprinted from reference 124, copyright Elsevier 2019. (**VI**) Nanobioconjugates based on single-wall carbon nanohorns doped with platinum nanoclusters (SWCNHs/PtNC) for the detection of α2,6-sialylated glycans, (A) preparation of SWCNHs/S-PtNC/M-APBA nanobioconjugates, (B) development of the electrochemical biosensing platform for target detection by using as-prepared nanobioconjugates, reprinted from reference 129, copyright Elsevier 2019.

**Figure 6 molecules-25-03542-f006:**
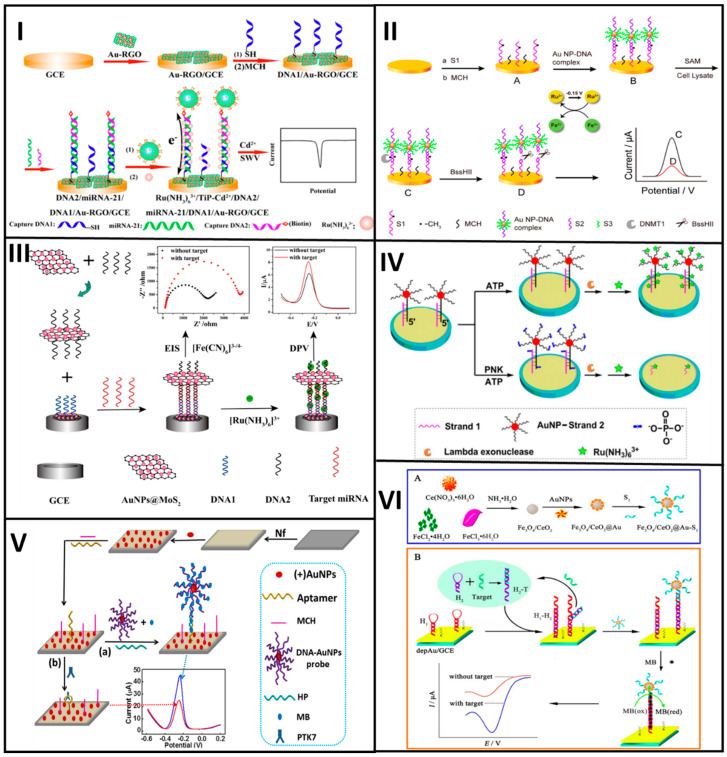
Stepwise schematic illustration of electrochemical biosensors based on electroactive complexes and dyes as signal amplifiers. (**I**) Biosensing platform to detect microRNA with nanobioconjugates based on streptavidin-Cd^2+^-modified titanium phosphate (TiP) nanoparticles as signal tags, reprinted from reference 133, copyright American Chemical Society 2015. (**II**) Electrochemical biosensor for the detection of microRNA-21 based on AuNPs functionalized with MoS_2_ signaling tags assisted by a ruthenium complex; (A) thiolated capture probe immobilization, (B) hybridization between capture probe and AuNP-DNA, (C) treatment of hybridized sequences with crude lysate containing *S*-adenosyl-l-methionine (SAM) (D) treatment of hybridized sequences with BssHII restriction endonuclease, reprinted from reference 139, copyright Elsevier 2017. (**III**) Electrochemical biosensor based on a DNA-AuNPs bioconjugate for the specific detection of DNA (cytosine-5)-methyltransferase 1 (DNMT1), reprinted from reference 140, copyright American Chemical Society 2016. (**IV**) Electrochemical biosensor based on DNA-functionalized AuNPs for the polynucleotide kinase (PNK) detection, reprinted from reference 141, copyright Elsevier 2017. (**V**) Electrochemical biosensor based on AuNP-coated Fe_3_O_4_/CeO_2_ hybrid material decorated with DNA for the detection of microRNA-21 (a) sandwich-type hybridization between aptamer, DNA-AuNPs and helper probe increasing MB catchment, (b) specific binding of PTK7 with aptamer limiting MB immobilization, reprinted from reference 142, copyright Elsevier 2018. (**VI**) Electrochemical biosensor based on AuNPs functionalized with ssDNA and assisted by MB, (A) development of hybrid nanomaterial Fe_3_O_4_/CeO_2_@Au and its functionalization with a DNA probe, (B) development of the electrochemical biosensing platform, reprinted from reference 143, copyright Elsevier 2016.

**Table 1 molecules-25-03542-t001:** Noble metal-based nanobioconjugates assembled at electrochemical biosensing platforms to detect different target biomolecules.

Nanobioconjugate	Target	Tech.	Linear Range	LOD	Ref.
DNA-AuNPs	*Sus scrofa* mitochondrial DNA	DPV	0.1–5.0 μg/mL	0.58 μg/mL	[24]
Aptamer-AgNCs	Cytochrome c	EIS	0.15–375 nM	72 pM	[25]
DNA-Pt@PdNCs	Pb^2+^	DPV	0.1–200 nM	0.033 pM	[26]
AuNPs@Dox@DNA	microRNA	EIS	1 pM–10 nM	0.17 pM	[27]
Aptamer-AgNPs	Mucin-1	DPASV	100–10^7^ cells/mL	25 cells/mL	[28]
DNA aptamer-PH-AuPd NPs	MCF-7 cells	DPV	50–10^7^ cells/mL	20 cells/mL	[29]
DNA-AuNPs	ORAOV1	PEC	1 fM–10 nM	0.33 fM	[31]
DNA-AuNPs	Micro-RNA-21	CV	5.0 fM–1.0 pM	1.6 fM	[32]
Antibody-AuNPs	Hepatitis B virus antigen	SWV	0.3–1000 pg/mL	0.19 pg/mL	[33]
Antibody-pillar[5]arene@AuNPs@g-C_3_N_4_	PSA	DPV	0.0005–10 ng/mL	0.12 pg/mL	[30]
Antibody-AgNPs	TBEV	CLSV	50–1600 IU/mL	50 IU/mL	[34]
Fc/AuNPs-strep	Circulating microRNA	DPV	10 fM–2 pM	5 pM	[35]

NPs, nanoparticles; NC, nanocluster; PH-AuPd, polyhedral AuPd; Fc, ferrocene; strep, streptavidin; ORAOV1, DNA oral cancer overexpressed 1; PSA, prostate-specific antigen; TBEV, tick-borne encephalitis; Tech., technique; SWV, square wave voltammetry; DPV, differential pulse voltammetry; CV, cyclic voltammetry; PEC, photoelectrochemical; EIS, electrochemistry impedance spectroscopy; CLSV, cathodic linear sweep voltammetry; DPASV, differential pulse anodic stripping voltammetry.

**Table 2 molecules-25-03542-t002:** Nanobioconjugates based on carbon allotropes, quantum dots, polymers, magnetic and oxide metal as signaling tags for the development of electrochemical biosensors.

Nanobioconjugate	Target	Detection Technique	Linear Range	LOD	Ref.
Antibody-CdSe/QDs	CEA	PEC	0.005–1000 ng/mL	5 pg/mL	[41]
DNA-QDs	BRCA1	DPV	5 aM–5 fM	1.2 aM	[42]
DNA apatamer-Concatemer/QDs	Cancer cell CCRF-CEM	ASV	100–10^6^ cells/mL	50 cells/mL	[43]
Aptamer-rGO/Th	Mucin-1	DPV	0.1–14.5 nM	0.06 nM	[52]
Poly-NTA-MWCNTs	Anti-cholera toxin	EIS	0.1 pg/mL–0.1 µg/mL	0.1 pg/mL	[53]
Antibody-SWCNTs	TGF-β1	Amperometry	2.5–1000 pg/mL	0.95 pg/mL	[54]
Antibody-Pt/Pd/Cu-GO	SCCA	EIS	0.0001–1 ng/mL	25 fg/mL	[55]
PY-rGO	CEA	EIS	0.1–1000 ng/mL	0.06 ng/mL	[56]
Antibody-CeO_2_/luminol	amyloid-β protein	ECL	80 fg/mL–100 ng/mL	52 fg/mL	[61]
Antibody-Au@CeO_2_	TSGF	EIS	0.5–100 pg/mL	0.2 pg/mL	[63]
Antibody-Fe3O4@SiO_2_	CEA	DPV	0.001–80 ng/mL	0.0002 ng/mL	[64]
Antibody-AuNPs/Fe_3_O_4_	CEA	CV	0.001–30 ng/mL	0.39 pg/mL	[65]
Antibody-Fe_3_O_4_	HER2	SWSV	0.5 pg/mL 50 ng/mL	0.02 pg/mL	[66]
Antibody-CeO_2_	HER2	CV	0.001–0.5 and 0.5–20 ng/mL	34.9 pg/mL	[67]
DNA aptamer-MNPs	Tumor exosomes	DPV	10^3^–1.2^5^ particles/µL	70 particles/µL	[68]
Antibody-PLLA	PSA and VEGF	DPV	0.05–100 and 1–100 ng/mL	50 pg/mL	[72]

QDs, quantum dots; rGO, reduced graphene oxide; Th, thionine; NPs, nanoparticles; MNPs, magnetic nanoparticles; PLLA, poly-lactide nanoparticles; PY, pyrenebutyric acid; CEA, carcinoembryonic antigen; BCRA1, microRNA of breast cancer 1 gene mutation; TGF-β1, transforming growth factor β1 cytokine; SCCA, squamous cell carcinoma antigen; TSGF, tumor-specific growth factor; HER2, human epidermal growth factor receptor 2; PSA, prostate-specific antigen; VEGF, vascular endothelial growth factor; PEC, photoelectrochemistry; DPV, differential pulse voltammetry; ASV, anodic stripping voltammetry; CV, cyclic voltammetry; SWSV, square wave stripping voltammetry.

**Table 3 molecules-25-03542-t003:** Nanobioconjugates employed as signal amplification tags in the development of electrochemical biosensing platforms.

Nanobioconjugate	Target	Technique	Linear Range	LOD	Reference
DNA-AuNPs	microRNA	DPV	0.1 fM–100 pM	0.05 fM	[100]
DNA-AuNP/WS_2_-AB	DNA	DPV	0.001–100 pM	0.12 pM	[105]
DNA-CdS:Mn/ZnS	HTLV-II	PEC	0.1–5000 fM	0.033 fM	[108]
Antibody-AuNRs-HRP	acrylamide	CV	187 ng/Kg–104 µg/Kg	56 ng/Kg	[113]
DNA/AuNPs, MWCNTs	DNA	DPV	10 fM–0.1 nM	2.5 fM	[112]
Antibody-HAP-ALP	BACE1	SWV	0.25–100 U/mL	0.1 U/mL	[114]
DNA-AuNPs/MoS_2_	microRNA	DPV	0.1 fM–0.1 pM	0.086 fM	[117]
DNA-AuNPs	microRNA	Chronocoulometry	0.01 fM–10 pM	6.8 aM	[86]
streptavidin/Th/Pt/Pd/CeO_2_	MMP-2	DPV	0.1 pg/mL–10 ng/mL	0.078 pg/mL	[123]
DNA-CdTe QDs	DNA	PEC	0.1 fM–5 pM	25.6 aM	[119]
Antibody-QDs/SiO_2_	Bcl-2 and Bax	ASV	1–250 ng/mL	0.5 fM	[125]
Fc/MPBA/AuNPs-SiO_2_	glycoprotein	DPV	1 pg/mL–250 ng/mL	0.57 pg/mL	[126]
DNA-AuNPs	microRNA-21	DPV	0.1 fM–0.1 nM	16 aM	[120]
DNA dendrimer	laminin (LN)	ECL	0.1 pg/mL–100 ng/mL	0.0661 pg/mL	[128]
M-APBA-SWCNH/Pt	α2,6-sialylated glycans	DPV	1 fg/mL–100 ng/mL	0.69 ng/mL	[129]
Aptamer-AuNPs/GO	Pb^2+^	Amperometry	5 pM–1µM	1.67 pM	[130]
DNA/HRP/Pt/CuMOF	microRNA-151	SWV	0.5–100 pM	0.13 fM	[121]
Aptamer-AuNPs/MoS_2_	ZEN and FB1 mycotoxins	DPV	1 µg/mL–(10)100 ng/mL	0.5 1 µg/mL	[131]
polyA-ODNs-AuNPs	Adenosine molecules	DPV	0.5 nM – 1–5 µM	0.17 nM	[132]
DNA-AuNPs	DNA	LSV	0.1 pM–10 nM	5.2 fM	[122]
Streptavidin/AuNPs	miRNA-21 detection	Amperometry	0.1 fM-100 pM	43.3 aM	[124]
Streptavidin/TiP-Cd^2+^	microRNA-21	SWV	1 aM–10 pM	0.76 aM	[133]
DNA-AuNPs/Ir	DNA	ECL	2 fM–1 pM	0.86 fM	[138]
DNA- AuNP@MoS_2_	microRNA-21	DPV, EIS	10 fM–1 nM	0.78 fM 0.45 fM	[140]
DNA-AuNPs	DNMT1	DPV	1–40 U/mL	0.3 U/mL	[139]
DNA-AuNPs	PNK	DPV	0.001–10 U/mL	7.8 × 10^−4^ U/mL	[141]
ssDNA/Fe_3_O_4_-CeO_2_@Au	microRNA-21	DPV	1 fM–1nM	0.33 fM	[143]
ssDNA/AuNPs	PTK-7	DPV	1–100 pM, 100 pM–1nM	372 fM	[142]
Antibody-Ru^3+^	GP73	ECL	15 pg/mL–0.7 ng/mL	15 pg/mL	[144]

WS2–AB, 2-dimensional tungsten disulfide–acetylene black; HAP, hydroxiapatite nanoparticles; ALP, Alkaline phosphatase; APBA, 3-aminophenyl boronic acid; polyA-ODNs, poly adenine oligonucleotides; HTLV-II, human T-cell lymphotropic virus type II; BACE1, β-site amyloid precursor protein cleaving enzyme 1; MMP-2, matrix metalloproteinase 2; Bcl-2, B-cell lymphoma 2; Bax, Bcl-2-associated X protein; ZEN: zearalenone; FB1, fumonisin B1; DNMT1, DNA (cytosine-5)-methyltransferase 1; PNK, polynucleotide kinase; GP73, Golgi protein 73; PTK-7, tyrosine kinase-7 protein; SWCNH, single-walled carbon nanohorns; MWCNT, multi-walled carbon nanotubes; MOF: metal organic frameworks; ECL, electrochemiluminescence.

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
