# Peer review of "Nanobioconjugates for Signal Amplification in Electrochemical Biosensing"

_molecules, 2020, doi:10.3390/molecules25153542_

Round 1

Reviewer 1 Report

This paper is really interesting for people involved in the development of electrochemical biosensors. The abstract is clear and concise. The review is well introduced, with clear outline and objectives.

One recurrent issue from my point of view is that all figures are  not cited in the right order in the text Please invert the citations or the paragraphs related to citations.Furthermore all figures are located in pages too far from their citation.

Figure 1a:

It is not clear for a non-specialist the link between NPs, biomolecules and target, ligands, shell layer and core. I recommend to explain in the text what is the link? For instance does the ligand correspond to the biomolecule? Do the shell layer and core correspond to the NPs?

Page 3, line 117:

What does mean GE?

Figure 2:

Figure 2IV is cited in the text before figure 2III. Please invert the citations or the paragraphs related to citations.

This figure is cited page 3 and is only visible page 7. It is too far from the citation.

Page 5, line 180:

The initials GO for graphene oxide are only introduced later page 8 line 328. It is pefereable to write here what does mean GO.

Page 5, line 225:

I guess that “anti-TBE” is indeed anti-TBEV? Is it true?

Page 5, lines 218-229 (reference 33):

This method was not develop to detect the virus TBE as it is said in the manuscript “By using antibody-based nanobioconjugates, Yekaterina Khristunova et. al., developed and electrochemical immunosensor based on AgNPs biofunctionalized with an antibody for the detection of tick-borne encephalitis (TBEV),”.

It was develop to detect antibodies against TBEV (see reference 33: “This study has shown that the obtained silver-labelled Ab@AgNP bioconjugates can be used in voltammetric immunoassays to determine antibodies to TBEV.”). Please modify the text.

Page 6, lines 261-263:

The authors wrote: “Along with the signal amplification achieved by the incorporation of nanobioconjugates at biosensors, the choice of the proper electrochemical technique to interrogate them is also crucial to achieving high sensitivity.”

This sentence is highlighting a very important aspect of electrochemical biosensor that needs to be developed here. I recommend to develop this aspect of the choice of the proper electrochemical technique, linked to the choice of nanoconjugates and especillay the choice of NPs.

Page 6, lines 266-273:

One information is lacking about QDs. Please give which elements are considered to be QDs? For instance CdSe-QDs. As for noble metal NPs, give some examples of QDs.

Page 7, lines 284:

Give the complete name for CdSe. It does not appear before.

Table 1:

This figure is cited page 6 and is only visible page 8. It is too far from the citation. Table 1 should be inserted within the pages dealing with noble metals NPs.

Figure 3:

Figure 3V and 3VI are cited in the text before figures 3I to 3IV. Why? Please invert the citations or the paragraphs related to citations.

This figure is cited for the first time page 7 and is only visible page 13. It is too far from the citation.

Page 8, lines 307:

Give the complete name for CdTe. It does not appear before.

Page 8, lines324:

What does mean CCRF-CEM?

Figure 4:

This figure is cited page 17 for the first time and is only visible page 20. It is too far from the citation.

Figure 5:

Again the different parts of figure 5 are cited in the text, but not in the order. Why? Please invert the citations or the paragraphs related to citations. Even figure 6 started to be cited while citations for figure 5 are not finished. For the reader it is confusing.

This figure is cited for the first time page 21 and is only visible page 25. It is too far from the citation.

Figure 6:

Again the different parts of figure 6 are cited in the text, but not in the order. Why? Please invert the citations or the paragraphs related to citations.

Reference 3:

  1. Nikolov, L.; Mamatarkova, V.; Slavchev, S.; Stoychev, S. The Convergence of Biotechnology and 1226 Nanotechnology as an Accelaeretaor of the Development of Biofilm Technologies. 2008, 36–47.

“Accelerator” wrongly spelling. The journal name is lacking.

Author Response

Reviewer 1

Comments and Suggestions for Authors

This paper is really interesting for people involved in the development of electrochemical biosensors. The abstract is clear and concise. The review is well introduced, with clear outline and objectives.

We thank the reviewer 1 for recognizing that our work represents a high contribution to the electrochemical biosensing field and for highlighting that it was well conducted.

  • One recurrent issue from my point of view is that all figures are not cited in the right order in the text Please invert the citations or the paragraphs related to citations. Furthermore all figures are located in pages too far from their citation.

We have taken into account the reviewer's concern and re-organized the figures and the references accordingly, as shown below.

  • Figure 1a:

It is not clear for a non-specialist the link between NPs, biomolecules and target, ligands, shell layer and core. I recommend to explain in the text what is the link? For instance does the ligand correspond to the biomolecule? Do the shell layer and core correspond to the NPs?

We have complemented the information related to Figure 1A and included a short text on page 2, lines 63-67

  • Page 3, line 117:

What does mean GE?

We have defined the acronym "GE" as 'gold electrode' on page 3, line 119.

  • Figure 2:

Figure 2IV is cited in the text before figure 2III. Please invert the citations or the paragraphs related to citations.

We have re-organized the text, and now Figure 2III is on page 5, line 197 and Figure 2IV is on page 6, line 212.

This figure is cited page 3 and is only visible page 7. It is too far from the citation.

We have moved the Figure 2 on page 4, line 139.

  • Page 5, line 180:

The initials GO for graphene oxide are only introduced later page 8 line 328. It is pefereable to write here what does mean GO.

We have defined the acronym GO earlier on page 2, line 57.

  • Page 5, line 225:

I guess that "anti-TBE" is indeed anti-TBEV? Is it true?

We agree with the reviewer; we have corrected the word on page 6, line 248.

  • Page 5, lines 218-229 (reference 33):

This method was not develop to detect the virus TBE as it is said in the manuscript "By using antibody-based nanobioconjugates, Yekaterina Khristunova et. al., developed and electrochemical immunosensor based on AgNPs biofunctionalized with an antibody for the detection of tick-borne encephalitis (TBEV),".

It was develop to detect antibodies against TBEV (see reference 33: "This study has shown that the obtained silver-labelled Ab@AgNP bioconjugates can be used in voltammetric immunoassays to determine antibodies to TBEV."). Please modify the text.

We agree with the reviewer. We confirm that the immunoassay was developed for the antibodies anti-TBEV detection and modified the text accordingly, describing the voltammetric immunoassay development. Please see page 6, lines 243-255.

  • Page 6, lines 261-263:

The authors wrote: "Along with the signal amplification achieved by the incorporation of nanobioconjugates at biosensors, the choice of the proper electrochemical technique to interrogate them is also crucial to achieving high sensitivity."

This sentence is highlighting a very important aspect of electrochemical biosensor that needs to be developed here. I recommend to develop this aspect of the choice of the proper electrochemical technique, linked to the choice of nanoconjugates and especillay the choice of NPs.

We have included a statement that complements our paragraph on page 7, lines 278-283.

  • Page 6, lines 266-273:

One information is lacking about QDs. Please give which elements are considered to be QDs? For instance CdSe-QDs. As for noble metal NPs, give some examples of QDs.

We have included a pair of statements including more information related to QDs on page 8, lines 294-297

  • Page 7, lines 284:

Give the complete name for CdSe. It does not appear before.

We have defined the acronym CdSe (cadmium selenide) on page 8, line 304.

  • Table 1:

This figure is cited page 6 and is only visible page 8. It is too far from the citation. Table 1 should be inserted within the pages dealing with noble metals NPs.

 We have moved Table 1 at the end of the noble metals NPs section, page 7, line 286.

  • Figure 3:

Figure 3V and 3VI are cited in the text before figures 3I to 3IV. Why? Please invert the citations or the paragraphs related to citations.

We have re-organized the figure putting the references in ascending order 3I, 3II, 3III, 3IV, see this on pages 8, line 305 and line 318; page 10, lines 414 and 434.

  • This figure is cited for the first time page 7 and is only visible page 13. It is too far from the citation.

We have moved Figure 3 at the end of the carbon nanobioconjugates section, page 11.

  • Page 8, lines 307:

Give the complete name for CdTe. It does not appear before.

We have defined the acronym CdTe (Cadmium telluride) on page 8, line 319.

  • Page 8, lines324:

What does mean CCRF-CEM?

We have defined the acronym CCRF-CEM on page 8, line 325-326. CCRF-CEM; A T lymphoblastoid line obtained from the peripheral blood of a 4-year-old Caucasian female with acute lymphoblastoid leukemia. We have included this information on pages 8, lines 336-337.

  • Figure 4:

This figure is cited page 17 for the first time and is only visible page 20. It is too far from the citation.

We have moved Figure 4 at the molecular strategies section, page 18.

  • Figure 5:

Again the different parts of figure 5 are cited in the text, but not in the order. Why? Please invert the citations or the paragraphs related to citations. Even figure 6 started to be cited while citations for figure 5 are not finished. For the reader it is confusing.

We have re-organized Figure 5 citation and put these in order 5I, 5II, 5III, 5IV, 5V and 5VI.

  • This figure is cited for the first time page 21 and is only visible page 25. It is too far from the citation.

We have moved Figure 5 on page 23.

  • Figure 6:

Again the different parts of figure 6 are cited in the text, but not in the order. Why? Please invert the citations or the paragraphs related to citations.

 We have re-organized Figure 6 and the citations.

  • Reference 3:
  1. Nikolov, L.; Mamatarkova, V.; Slavchev, S.; Stoychev, S. The Convergence of Biotechnology and 1226 Nanotechnology as an Accelaeretaor of the Development of Biofilm Technologies. 2008, 36–47.

"Accelerator" wrongly spelling. The journal name is lacking.

 We have properly cited reference 3, including the journal on page 32, lines 1277-1279.

Reviewer 2 Report

The authors reviewed recent progress on nanoconjugates for electrochemical biosensors. The manuscript presents representative examples of use of various nanoconjugates and signal amplification methods with different nanomaterials as part of biohybrid materials. In general, the topic of the manuscript is appropriate to the review for biosynthesis of nanoparticles. Considering that Molecules publishes considerably reviews and novel studies for in-depth understanding of materials, I would recommend the publication of this manuscript on Molecules. The minor revision of the manuscript is required to improve the quality of the manuscript. Following suggestions and concerns should be clearly addressed.

In page 3 figure 1B, the number of symbols is different from that of the listed biomolecules in the figure caption. Also, suggesting placing the labels on the figure 1B.

In page 6, line 266, there are a few studies reviewed in this chapter. The authors could’ve included more studies for the QD based nanobioconjugates.

In page 21, lines 811-812, the statement should be corrected.

In page 22, line 863, the figure number is missed.

Author Response

Reviewer 2

Comments and Suggestions for Authors

The authors reviewed recent progress on nanoconjugates for electrochemical biosensors. The manuscript presents representative examples of use of various nanoconjugates and signal amplification methods with different nanomaterials as part of biohybrid materials. In general, the topic of the manuscript is appropriate to the review for biosynthesis of nanoparticles. Considering that Molecules publishes considerably reviews and novel studies for in-depth understanding of materials, I would recommend the publication of this manuscript on Molecules. The minor revision of the manuscript is required to improve the quality of the manuscript. Following suggestions and concerns should be clearly addressed.

We thank the reviewer for recognizing our efforts in the review writing and appreciate the reviewer's recommendations to improve the manuscript.

  • In page 3 figure 1B, the number of symbols is different from that of the listed biomolecules in the figure caption. Also, suggesting placing the labels on the figure 1B.

We have listed in the figure caption all illustrated biomolecules of Figure 1B on page 3, line 97-98. Additionally, we have organized the figure and included the labels on this figure.

  • In page 6, line 266, there are a few studies reviewed in this chapter. The authors could've included more studies for the QD based nanobioconjugates.

We have included additional studies based on QD nanobioconjugates for electrochemical biosensors development on pages 8-9, lines 340-379.

  • In page 21, lines 811-812, the statement should be corrected.

We have changed the statement "They not only enjoy enhanced physical, chemical and electrochemical properties but can be resistant to corrosion and oxidation processes" by "They not only have enhanced physical, chemical and electrochemical properties but are resistant to corrosion and oxidation processes and environments" on page 21, line 862-863.

  • In page 22, line 863, the figure number is missed.

We have organized the text and we have put the figure number (Figure 5III) on page 23, line 913.